# GEOMETRIC LAPLACE NEURAL OPERATOR

## ABSTRACT

Neural operators have emerged as powerful tools for learning mappings between function spaces, enabling efficient solutions to partial differential equations across varying inputs and domains. Despite the success, existing methods often struggle with non-periodic excitations, transient responses, and signals defined on irregular or non-Euclidean geometries. To address this, we propose a generalized operator learning framework based on a pole–residue decomposition enriched with exponential basis functions, enabling expressive modeling of aperiodic and decaying dynamics. Building on this formulation, we introduce the Geometric Laplace Neural Operator (GLNO), which embeds the Laplace spectral representation into the eigen-basis of the Laplace–Beltrami operator, extending operator learning to arbitrary Riemannian manifolds without requiring periodicity or uniform grids. We further design a grid-invariant network architecture (GLNONet) that realizes GLNO in practice. Extensive experiments on PDEs/ODEs and real-world datasets demonstrate our robust performance over other state-of-the-art models.

## 1 INTRODUCTION

Modern science and engineering applications increasingly demand efficient and adaptive simulations, from aerodynamics and robotics to climate and biological surface modeling. These systems are governed by ordinary or partial differential equations (ODEs/PDEs) describing underlying physical laws, characterized by evolving boundaries, heterogeneous parameters, and geometric complexity. Neural operators have emerged as a powerful tool for learning mappings between infinite-dimensional function spaces, enabling efficient approximation of PDE solutions across inputs and geometries (Kovachki et al., 2023; 2024; Azizzadenesheli et al., 2024). Unlike traditional solvers, they bypass repeated discretization, offering scalability and flexibility for data-driven modeling.

Earlier frameworks, such as DeepONet (Lu et al., 2021), leverage the universal approximation theorem for operators, while the Fourier Neural Operator (FNO) (Li et al., 2020a) performs global convolution in Fourier space. A family of successors (Wen et al., 2022; Pathak et al., 2022; Raonic et al., 2023; Liu et al., 2023) builds on FNO with targeted enhancements to accuracy, regularity across scales, generalization across domains, and handling of multiscale physics. Recent efforts adapt FNO to the sphere by replacing fast Fourier transform (FFT) with spherical harmonics transform (Bonev et al., 2023; Lin et al., 2023; Mahesh et al., 2024a;b; Hu et al., 2025). However, these methods remain limited to structured manifolds, e.g., spheres or tori, where symmetry and spectral tools are well-defined. As a result, they are not applicable to non-Euclidean domains with irregular topology or complex boundary conditions, common in real-world data modeling such as anatomical surfaces, geophysical data, or biological shapes. Other efforts extend FNO to complex geometries (Li et al., 2023a; Duprez et al., 2025). While effective for quasi-periodic signals on fixed meshes, these methods inherit the core limitations of Fourier analysis in representing non-periodic excitations and transient regimes that decay or grow exponentially (Cao et al., 2024).

The Laplace domain offers a potential alternative with generalizable representations. Recent work, such as the Laplace Neural Operator (LNO) (Cao et al., 2024), proposes a pole-residue formulation for learning transient responses in time-dependent systems. However, such formulations still rely on periodicity assumptions in the spectral transform (Cao et al., 2024), and uniform grid-structured domains, limiting their applicability to real-world systems defined on irregular meshes or curved surfaces. Separately, graph-based and transformer-based neural operators (Hao et al., 2023; Wen et al., 2025; Sarkar & Chakraborty, 2025a; Li et al., 2025) have been developed for arbitrary manifolds. However, these operators face high computational costs in manifold spaces and offer limited in-

terpretability, particularly for systems with distinct steady-state and transient components. Overall, existing operators remain limited in expressiveness for aperiodic, transient signals and generalizability to complex geometries, which hinders their applicability to truly aperiodic or decaying systems and data defined on irregular meshes or non-Euclidean domains.

**Our Approach. (1) Theoretical formulation**: To address this, we propose a generalized Laplace Basis & Transform that incorporates a learnable exponential basis, enabling expressive modeling of non-periodic and decaying signals. We rigorously formulate this basis to enhance approximation capability while maintaining computational feasibility. **(2) Neural operator**: To adapt this framework to non-Euclidean domains, we introduce the **Geometric Laplace Neural Operator (GLNO)**, a novel spectral learning method for Riemannian manifolds, which lifts the pole-residue calculus from the complex plane to the Laplace-Beltrami eigen-basis. This enables an end-to-end learnable spectral operator intrinsically aligned with the geometry while modeling aperiodic and transient signals, for principled learning on curved and irregular surfaces. **(3) Network architecture**: We implement a grid-invariant architecture that realizes GLNO with stackable operator layers. This architecture supports flexible discretizations and strong generalization across geometries and transient dynamics, particularly suited for applications such as biophysical simulations on anatomical surfaces, climate forecasting over irregular terrains, and structural learning in molecular biology.

Our contribution includes:

- **Generalized Laplace Basis & Transform** for operator learning, with expressive modeling of non-periodic, decaying signal modeling through exponential bases.

- **Geometric Laplace Neural Operator (GLNO)** that generalizes Laplace-based operator learning to arbitrary Riemannian manifolds via the Laplace-Beltrami operator.

- **Grid-invariant network architecture** based on GLNO, enabling generalized learning on arbitrary manifolds and geometric feature fusing.

- **Extensive experiments** on regular-grid and geometric PDEs, and real-world tasks, showing superior accuracy, generalization, and parameter efficiency over state-of-the-art models.

## 2    RELATED WORK

Extending neural operators to complex geometries has been central for solving PDEs on non-Euclidean domains. Early approaches, e.g., Geo-FNO and $\phi$-FEM-FNO, map irregular geometries onto regular grids, to leverage FFT-based kernel from FNO (Li et al., 2023a; Duprez et al., 2025). Following work eliminates FFT entirely and defines spectral convolution directly on the manifold using Laplace–Beltrami operator (LBO) (Chen et al., 2023a). **Transformer-based approaches** integrate geometry-aware features—extracted from irregular point clouds or meshes—into neural operators, extending to arbitrary geometries without relying on fixed grids (Hao et al., 2023; Liu et al., 2025; Fei et al., 2025; Han et al., 2025; Shi et al., 2025; Ramezankhani et al., 2025; Wen et al., 2025). However, they often suffer from high computational costs (Hao et al., 2023; Wen et al., 2025), especially for fine-resolution meshes. **Graph-based neural operators** offer another strategy, modeling parametric integral kernels via message passing (Li et al., 2020b;c; Sarkar & Chakraborty, 2025a; Li et al., 2025). However, they face challenges of parameter complexity and computational costs due to localized operations and graph construction. Other research has explored LBO for geometric spectral learning methods (Liang et al., 2012; Smirnov & Solomon, 2021; Sharp et al., 2022; Wiersma et al., 2022; Chen et al., 2023b), including diffusion-based spectral encoding achieving high performance and efficiency in diverse scenarios (Sharp et al., 2022; Zhu et al., 2024; 2025)

## 3    PRELIMINARY

### 3.1    LAPLACE NEURAL OPERATOR

The Laplace Neural Operator (LNO) Cao et al. (2024) provides a foundational framework for learning operators in the complex spectral domain through pole-residue formulations.

The input function $v(t)$ is decomposed into Fourier series, $f(x) = \sum_\omega \alpha_\omega e^{i\omega x}$, and transformed into Laplace spectral domain using Laplace transform:

$$F(s) = \mathcal{L}\{f(x)\} = \alpha_\omega \sum_\omega \mathcal{L}\{e^{i\omega t}\} = \sum_\omega \frac{\alpha_\omega}{s - i\omega} \tag{1}$$

where the coefficients $\alpha_\omega$ can be efficiently computed via FFT.

LNO parameterized the operator kernel directly in the Laplace complex domain using a pole-residue formulation (the complete mathematical derivation in Appendix B.1):

$$K_\theta(s) = \sum_{n=1}^N \frac{\beta_n}{s - \mu_n}, \quad \mu_n \in \mathbb{C}, \beta_n \in \mathbb{C} \tag{2}$$

where $\{\mu_n\}$ are learnable poles and $\{\beta_n\}$ are corresponding learnable residues.

The convolution operator in Laplace domain is performed through multiplication:

$$G(s) = K_\theta(s) \cdot F(s) = \Big( \sum_{n=1}^N \frac{\beta_n}{s - \mu_n} \Big)\Big( \sum_\omega \frac{\alpha_\omega}{s - i\omega} \Big) = \sum_{n=1}^N \frac{\hat{a}_n^{\text{transient}}}{s - \mu_n} + \sum_\omega \frac{\hat{a}_\omega^{\text{steady}}}{s - i\omega} \tag{3}$$

where the residues are given by residue calculus (the complete mathematical derivation in Appendix B.2):

$$\hat{a}_n^{\text{transient}} = \lim_{s \to \mu_n} (s - \mu_n)G(s) = \beta_n F(\mu_n), \quad \hat{a}_\omega^{\text{steady}} = \lim_{s \to i\omega} (s - i\omega)G(i\omega) = \alpha_\omega K_\theta(i\omega) \tag{4}$$

The output reconstruction employs inverse Laplace transformation:

$$g(t) = \mathcal{L}^{-1}\{G(s)\} = \sum_{n=1}^N \hat{a}_n^{\text{transient}} e^{\mu_n t} + \sum_\omega \hat{a}_\omega^{\text{steady}} e^{i\omega t} \tag{5}$$

## 3.2 LAPLACE-BELTRAMI OPERATOR AND SPECTRAL GEOMETRY

Let $\Delta_\mu$ be the Laplace-Beltrami operator (Bobenko & Springborn, 2007) (Boscain & Laurent, 2013) on a Riemannian manifold $(\mathcal{M}, \mu)$, where $\mathcal{M}$ is a compact, connected manifold and $\mu$ represents the Riemannian volume element $\mathrm{d}\mu$ with the inner product on $L^2(\mathcal{M})$ $\langle f, g \rangle_\mu := \int_{\mathcal{M}} f(x)g(x)\,\mathrm{d}\mu(x)$ . The Laplace-Beltrami operator admits a discrete spectrum with countable, ordered non-negative eigenvalues $\lambda_k$ and corresponding eigenfrequency $\omega_k = \sqrt{\lambda_k}$ and a family of orthonormal eigenfunction $\phi_{\omega_k}$:

$$-\Delta_\mu \phi_{\omega_k} = \lambda_k \phi_{\omega_k} = \omega_k^2 \phi_{\omega_k}, \quad 0 = \omega_1 < \omega_2 \le \omega_3 \le \cdots, \quad \langle \phi_\omega, \phi_{\omega'} \rangle_\mu = \delta_{\omega\omega'} \tag{6}$$

## 4 METHODOLOGY

### 4.1 GENERALIZED LAPLACE NEURAL OPERATOR WITH NON-PERIOD BASIS

Conventional spectral operators, such as the Fourier Neural Operator (FNO) and Laplace Neural Operator (LNO), rely on a basis of pure complex exponentials, $\{e^{-i\omega t}\}$, which inherently assumes periodicity and is limited in modeling transient or decaying phenomena. To address this, we introduce a **generalized Laplace basis** that provides a more versatile spectral representation.

**Generalized Laplace Basis in Spectral Domain.** We define our basis in the complex domain as:

$$\{\varepsilon_z(x)\} = \{e^{-zx}\} = \Big\{ \underbrace{e^{-\sigma x}}_{\text{non-periodic}} \cdot \underbrace{e^{-i\omega x}}_{\text{periodic}} \Big\}, \quad z = \sigma + i\omega \in \mathbb{C} \tag{7}$$

where spectral coordinates $z$ encode both the frequency ($\omega = \mathrm{Im}(z)$) and the temporal characteristics ($\sigma = \mathrm{Re}(z)$) of the signal. The inclusion of the real component $\sigma$ enables the representation of exponential decay or growth phenomena at different scales, in addition to oscillatory behavior. This forms a strict superset of the Fourier basis to enhance representation of transient dynamics, which is recovered when $\sigma = 0$.

**Spectral Decomposition.** Considering the inner product for functions in the time domain as $\langle f(x), g(x) \rangle := \int_0^\infty f(x)\overline{g(x)}dx$, we define the decomposition for any function $f(x)$ as:

$$\mathcal{D}\{f(x)\}(z) = \langle f, \varepsilon_z \rangle \varepsilon_z(x) = \left( \int_0^\infty f(x)e^{-\sigma x}e^{i\omega x}dx \right)\varepsilon_z(x) = \langle e^{-\sigma x}f, e^{-i\omega x}\rangle \varepsilon_z(x) \quad (8)$$

where the coefficients $\langle e^{-\sigma x}f, e^{-i\omega x}\rangle$ can be efficiently computed via FFT for function $e^{-\sigma x}f$. This efficient coefficient computation maintains the operator's computational feasibility while significantly increasing its representation for complex physical phenomena.

**Neural Operator with the Non-period Basis.** Following the LNO framework in Section 3.1, we apply the decomposition with the generalized Laplace basis to the Laplace transform. The Laplace transform of the basis function $\varepsilon_z(t)$ is computed as:

$$\mathcal{L}\{\varepsilon_z(t)\} = \int_0^\infty e^{-zt}e^{-st}dt = \frac{1}{s + z} \quad (9)$$

In practical implementation, we approximate the decomposition using learnable parameters $\{z_i\}$ and denotes the number as $M$:

$$\mathcal{D}\{f(t)\} = \sum_{i=1}^M \alpha_i\varepsilon_{z_i}(t), \quad \alpha_i = \langle f, \varepsilon_{z_i}\rangle, \quad z_i = \sigma_i + i\omega_i \quad (10)$$

This decomposition is then applied within the LNO framework for Laplace transformation and the convolution using pole-residue form and the output calculated through inverse Laplace transformation:

$$g(t) = \mathcal{L}^{-1}\{\mathcal{L}\{\mathcal{D}\{f\}\}K_\theta\} = \sum_{n=1}^N \hat{a}_n^{\text{transient}}e^{\mu_n t} + \sum_{i=1}^M \hat{a}_i^{\text{steady}}e^{-z_i t} \quad (11)$$

where $K_\theta$ is the kernel in Laplace domain in pole-residue form. Detailed mathematical procedures are provided in Appendix B.3.

## 4.2 GEOMETRIC LAPLACE NEURAL OPERATOR (GLNO)

Building upon the generalized Laplace framework for Euclidean domain, we now extend this approach to arbitrary manifolds and unstructured meshes. The central challenge—and our primary contribution in this work—lies in constructing a basis intrinsically defined by the manifold's structure and extending the Laplace Transform framework to geometric domains.

**Geometric Laplace Basis and Decomposition.** We construct a basis for $L^2(\mathcal{M})$ by synthesizing the spectral geometry of the manifold with our generalized Laplace basis (Section 4.1):

$$\{\varepsilon_z(x)\} = \Big\{ \underbrace{\exp(-\sigma\mathcal{P}(x))}_{\text{non-periodic}} \cdot \underbrace{\phi_\omega(x)}_{\text{periodic}} \Big\}, \quad z = \sigma + i\omega_k \in D \subset \mathbb{C} \quad (12)$$

where $\mathcal{P} : \mathcal{M} \to \mathbb{R}$ encodes intrinsic geometric properties (e.g., curvature), $\phi_\omega$ are the eigenfunctions of the Laplace-Beltrami operator with eigenfrequency $\omega$, given in Section 3.2, and $D$ denotes the discrete spectral domain determined by the manifold's geometry.

The decomposition of any function $f \in L^2(\mathcal{M})$ is given by approximation with learnable parameters $\{z_i\}_{i=1}^M$:

$$f(x) = \sum_{i=1}^M \alpha_i\varepsilon_{z_i}(x) = \sum_{i=1}^M \langle \exp(-\sigma_i\mathcal{P})f, \phi_{\omega_i}\rangle_\mu \varepsilon_{z_i}(x) \quad (13)$$

where $\alpha_z$ denotes the spectral coefficient computed via the inner product on $\mathcal{M}$.

**Geometric Laplace Transform.** We establish a unified mapping from geometric basis functions to the complex spectral domain as the geometric Laplace transform (the complete mathematical derivation in Appendix B.4):

$$\mathcal{L}\{\varepsilon_z(x)\} \mapsto \frac{1}{s + z}, \quad \forall z \in D \quad (14)$$

This mapping preserves the physical interpretation of both oscillatory ($\omega$) and decay/growth ($\sigma$) components while adapting them to the manifold's intrinsic structure.

Due to the completeness of $\{\phi_\omega\}$, every meromorphic residue term can be exactly synthesized by the same eigen-basis, providing a uniform representation for arbitrary geometries that enables operator learning across different manifold shapes.

**Operator Action in Laplace Domain.** Following the LNO framework, we parameterize the operator kernel in pole-residue form $K_\theta(s) = \sum_{n=1}^{N} \frac{\beta_n}{s - \mu_n}$ with learnable parameters $\{\mu_n\}$ and $\{\beta_n\}$. (The detailed theorem is given in Appendix B.4)

The operator action is performed through multiplication in the spectral domain:

$$G(s) = F(s)K_\theta(s) = \left( \sum_{i=1}^{M} \frac{\alpha_i}{s + z_i} \right) \left( \sum_{n=1}^{N} \frac{\beta_n}{s - \mu_n} \right) \tag{15}$$

Using residue calculus, the output is decomposed into steady-state and transient components:

$$G(s) = \sum_{i=1}^{M} \frac{\hat{a}_i^{\text{steady}}}{s + z} + \sum_{n=1}^{N} \frac{\hat{a}_n^{\text{transient}}}{s - \mu_n} \tag{16}$$

where the residues are calculated similar to Equation 4.

**Inverse Geometric Laplace Transform.** The inverse transform reconstructs the output function on the manifold. A fundamental challenge arises because learned kernel poles $\mu_n$ are continuous complex poles that may not align with the discrete spectral coordinates $z \in D$.

Our solution employs a Gaussian filter to project continuous poles $\mu_n$ to discrete pole domain $D$:

$$Gauss(x) = \frac{1}{\sqrt{2\pi}\Sigma} \exp\left( -\frac{x^2}{2\Sigma^2} \right) \tag{17}$$

$$\mathcal{L}^{-1} \left\{ \frac{1}{s - \mu_n} \right\} = \exp(\text{Re}(\mu_n)\mathcal{P}) \cdot \sum_\omega Gauss(\text{Im}(\mu_n) - \omega)\phi_\omega(x) \tag{18}$$

This inverse Laplace transform preserves the geometric interpretation of $\text{Re}(\mu_n)$ as controlling decay/growth via $\mathcal{P}(x)$ while mapping the continuous frequency $\text{Im}(\mu_n)$ to the discrete spectral basis $\{\phi_\omega\}$ through smooth interpolation. This approach maintains the expressive power of continuous spectral learning while respecting the discrete nature of manifold spectra.

The final reconstruction combines both components:

$$g(x) = \mathcal{L}^{-1}\{G(s)\}(x) = \sum_{i=1}^{M} \hat{a}_i^{\text{steady}} \varepsilon_{z_i}(x) + \sum_{n=1}^{N} \hat{a}_n^{\text{transient}} \cdot \mathcal{L}^{-1} \left\{ \frac{1}{s - \mu_n} \right\} \tag{19}$$

### 4.3 GEOMETRIC LAPLACE NEURAL OPERATOR NETWORK

Based on the mathematical formulation presented above, we design the Geometric Laplace Neural Operator block and overall GLNO architecture to learn mappings between function spaces defined on manifolds, shown in Figure 1. Specifically, given an input function $f : \mathcal{M} \to \mathbb{R}^{d_{in}}$ and a target output function $u : \mathcal{M} \to \mathbb{R}^{d_{out}}$, GLNO learns the operator $\mathcal{G} : f \mapsto u$ through a spectral approach that inherently respects the underlying geometric structure.

The overall GLNO architecture processes geometric data through a pipeline that begins with computing intrinsic geometric features (curvature, boundary distances) and Laplace-Beltrami eigen-pairs using SciPy (Virtanen et al., 2020), followed by an encoding MLP $\mathcal{P} : \mathbb{R}^{d_{in}} \to \mathbb{R}^{d_{latent}}$. The core of the network consists of multiple GLNO blocks that sequentially transform these features, culminating in a final projection $\mathcal{Q} : \mathbb{R}^{d_{latent}} \to \mathbb{R}^{d_{out}}$.

For the implementation of each GLNO block shown in Figure 1, a learnable decomposition on the input manifold is calculated and the pole and residue value is obtained. Then, the Laplace transform and pole residue calculation is performed according to Section 19. Each GLNO block integrates

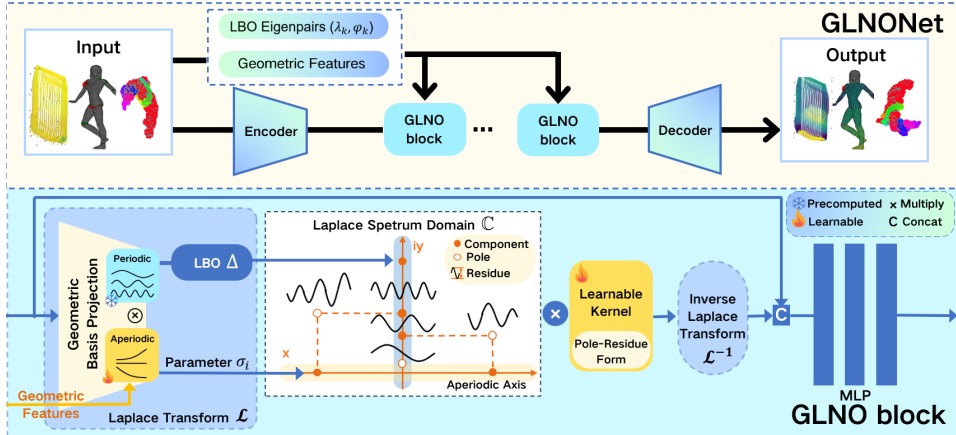

Figure 1: GLNONet & GLNO block architectures. Encoder and Decoder are both multi-layer perceptron (MLP). In GLNO block, geometric features include coordinate, curvature and distance. GLNO achieves spectral learning of transient response through pole-residue calculation in manifolds and learnable frequency domain transform ($\sigma_i$). LBO: Laplace-Beltrami operator.

the transformed features with original geometric descriptors through lightweight MLPs and skip connections.

The grid invariance of GLNO stems from its foundation on the Laplace-Beltrami operator, whose eigenfunctions provide a coordinate-free representation of the manifold that is independent of any particular discretization and connection, without the need of re-training, re-sampling or inserting dummy vertices. The incorporation of geometric features such as curvature further enhances this invariance while providing crucial contextual information about local manifold structure, enabling the network to adaptively weight spectral components based on intrinsic geometric properties rather than relying on potentially inconsistent coordinate representations.

## 5 EXPERIMENTS & RESULTS

Table 1: Left: Parameters and description for every benchmarks; Right: Algorithm analysis for baseline models

| Task | Mesh Type | Benchmarks | Geometry | dim | Vertices |
|---|---|---|---|---|---|
| ODEs/PDEs | Fixed | Duffing | Structured | 1 | 2048 |
| | | Lorenz | Structured | 1 | 2048 |
| | | Pendulum | Structured | 1 | 2048 |
| | | Beam | Structured | 2 | 50×50 |
| | | Diffusion | Structured | 2 | 50×50 |
| | | Reaction-Diffusion | Structured | 2 | 40×20 |
| | | Poisson | Unstructured | 2 | 3242 |
| Simulation | Dynamic | Scape-Net Car | Unstructured | 3 | 32186 |
| Classification | | SHREC-11 | Unstructured | 3 | 5000 |
| Classification | | RNA | Unstructured | 3 | 10000 |
| Classification | | Human | Unstructured | 3 | 10000 |

| Models | Unstructured | Fourier-Base | Transformer-Base | Graph-Base |
|---|---|---|---|---|
| WNO | ✗ | ✗ | ✗ | ✗ |
| FNO | ✗ | ✓ | ✗ | ✗ |
| CNO | ✗ | ✗ | ✗ | ✗ |
| LNO | ✗ | ✗ | ✗ | ✗ |
| Geo-FNO | ✓ | ✓ | ✗ | ✗ |
| GINO | ✓ | ✓ | ✗ | ✓ |
| LSM | ✓ | ✓ | ✗ | ✗ |
| GKNO | ✓ | ✗ | ✗ | ✗ |
| Sp2GNO | ✓ | ✓ | ✗ | ✓ |
| GNOT | ✓ | ✗ | ✓ | ✗ |
| Transolver | ✓ | ✗ | ✓ | ✗ |
| AMG | ✓ | ✗ | ✓ | ✓ |

We conduct a comprehensive evaluation to assess the performance of the proposed GLNO across diverse and challenging scenarios. The experiments are purposefully structured into two main categories:

**First, on ODEs/PDEs with fixed meshes**, we aim to evaluate the operator's fundamental capabilities, including learning periodic responses (e.g., Pendulum with $c = 0$), modeling chaotic systems (e.g., Lorenz system), reacting to nonlinear signals (e.g., Pendulum with $c > 0$), and handling problems in higher dimensions (e.g., 2D Beam and Diffusion). We include tasks on unstructured meshes (Poisson equation) to test the model's flexibility and performance beyond regular grids.

**Second, on real-world simulations and classifications with dynamic and unstructured meshes (Scape-Net Car, RNA, Human Body)**, our goal is to validate the model's generalization ability and practical utility in complex, real-world settings.

To ensure a rigorous and fair comparison, we benchmark our model against a wide range of neural operators. As detailed in the table 1, these baselines are selected to represent different architectural paradigms, including Fourier-based (e.g., FNO, Geo-FNO), Transformer-based (e.g., GNOT, Transolver), and Graph-based (e.g., GKNO) methods. All experiments are implemented through PyTorch on 32GB V100 GPUs. More experimental details, including parameters and baseline fine-tuning can be found in Appendix A.

## 5.1 LEARNING NONLINEAR AND NON-PERIODIC RESPONSES ON ODEs/PDEs

| Problem | Parameters | Governing Equation | CNO | FNO | WNO | LNO | | GLNO (Ours) | |
|---|---|---|---|---|---|---|---|---|---|
| **Blocks** | | | 4 | 4 | 4 | 1 | 4 | 1 | 4 |
| **Channels** | | | 64 | 64 | 64 | 8 | 8 | 8 | 8 |
| Driven Pendulum | $c = 0$ | $f(t) = \ddot{x} + c\dot{x} + \sin(x)$ | 0.6268 | 0.3668 | 0.6090 | 0.8461 | 0.9997 | 0.6752 | **0.2916** |
| | $c = 0.5$ | | 0.1540 | 0.1718 | 0.0965 | 0.1420 | 0.9885 | 0.1682 | **0.0875** |
| Duffing Oscillator | $c = 0$ | $f(t) = \ddot{x} + c\dot{x} + x + x^3$ | 0.7885 | 0.4681 | 0.7607 | 0.9157 | 0.9998 | 0.8926 | **0.4416** |
| | $c = 0.5$ | | 0.1424 | 0.1362 | 0.0987 | 0.8347 | 0.9747 | 0.4169 | **0.0725** |
| Lorenz System | $\rho = 5$ | $\dot{x} = 10(y - x), \dot{y} = x(\rho - z) - y$ | **0.0051** | 0.0185 | 0.0130 | 0.1071 | 0.1133 | 0.0368 | 0.0240 |
| | $\rho = 10$ | $\dot{z} = xy - \frac{8}{3}z - f(t)$ | 0.4818 | 0.5050 | 0.4924 | 0.5833 | 0.4323 | 0.2481 | **0.2187** |
| Beam Equation | - | $f(x,t) = EI\frac{\partial^4 w}{\partial x^4} + \rho A\frac{\partial^2 w}{\partial t^2}$ | 0.0293 | 0.0034 | 0.1511 | 0.0083 | 0.3917 | 0.0094 | **0.0026** |
| Diffusion Equation | D=1 | $f(x,t) = D\frac{\partial^2 y}{\partial x^2} - \frac{\partial y}{\partial t}$ | 0.1143 | 0.0064 | 0.0237 | 0.0011 | 0.1392 | **0.0006** | 0.0024 |
| Reaction-Diffusion | D=0.01,k=0.01 | $f(x,t) = D\frac{\partial^2 y}{\partial x^2} + ky^2 - \frac{\partial y}{\partial t}$ | - | 0.0909 | 0.1909 | 0.1278 | 0.2852 | **0.1014** | 0.1821 |

Table 2: Performance comparison on structured grid ODEs/PDEs problems (Relative Error). **Bold** indicates the best result for each problem.

**Experimental Configuration and Parameters:** Specific information for driven function $f(t)$ in different tasks is illustrated in Appendix A.

All models are implemented under consistent conditions using the Adam optimizer with an initial learning rate of 0.001. A step decay schedule is applied, reducing the learning rate by half every 100 epochs. Training proceeded for 1,000 epochs with validation every 10 epochs.

CNO (Raonic et al., 2023) is included to compare spectral methods with non-spectral neural operator approaches. FNO and LNO are selected to evaluate the impact of employing non-periodic basis functions, while WNO (Navaneeth et al., 2024), which utilizes wavelet methods, serves as a benchmark for comparing the capability to handle non-periodic phenomena.

**Spectral *vs.* Non-Spectral Method Analysis:** Across the majority of benchmark problems, spectral operators such as FNO and GLNO consistently outperform non-spectral methods, highlighting the advantage of global basis representations in capturing long-term and global dependencies for operator learning in ODEs and PDEs. Among the spectral approaches, GLNO achieves the best or comparable performance, demonstrating the effectiveness of our generalized aperiodic spectral formulation. The only notable exception occurs in the chaotic Lorenz system with $\rho = 0.5$, where the non-spectral CNO attains the lowest error (0.0051 relative error). This can be attributed to the highly localized and chaotic dynamics of the Lorenz system, which are more suitably represented by CNO's local inductive bias—further underscoring that the choice of spectral versus non-spectral methods should be guided by the nature of the underlying system.

**Non-periodic *vs.* Periodic Handling Capability:** GLNO achieves superior performance in problems involving damping and transient behaviors, especially in pendulum systems with varying damping coefficients (Fig. 2). This advantage demonstrates our generalized Laplace basis's abil-

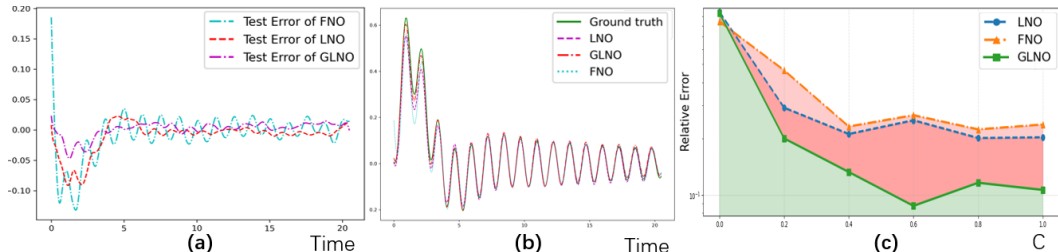

Figure 2: Driven Gravity Pendulum Result: (a) The test Error of different models in $c = 1.0$ pendulum task; (b) The prediction of different models in $c = 1.0$ pendulum task; (c) The Relative Error$L^2$-$c$ curve for different models.

ity to capture both exponential decay/growth patterns and oscillatory behaviors, whereas traditional spectral methods are limited.

**Architecture Scalability and Parameter Efficiency.** LNO-4-blocks exhibits degraded performance across multiple tasks, while GLNO-4-block maintains robust performance, demonstrating its superior architectural scalability. This divergence can be attributed to LNO's sensitivity to training configurations—under our unified training schedule, LNO struggles to converge optimally. In contrast, GLNO's structurally efficient design enables stable depth scaling without compromising performance, particularly evident in 1D problem settings.

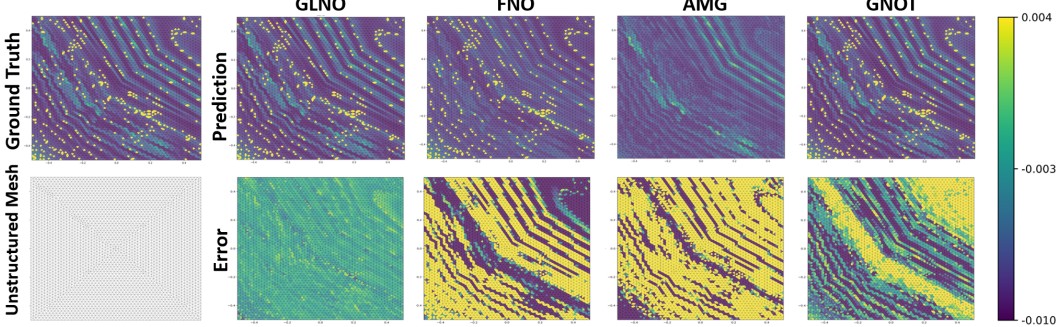

Figure 3: Poisson Equation results from different models and unstructured mesh structure used in this task

**Poisson Equation on Unstructured Meshes.** To validate the generalization of our method beyond structured grids, we evaluate on the Poisson equation defined on unstructured meshes over a rectangular domain, following the setup of (Li et al., 2025):

$$-\Delta u = f, \text{in} \, \Omega = [0,1]^2$$
$$u = 0, \text{on} \, \partial\Omega \qquad (20)$$

GLNO learns the operator mapping from source terms $f(x, y)$ to solutions $u(x, y)$ on these unstructured discretizations. The training set contains 4000 samples with Gaussian source terms parameterized by $\mu_{x,i}, \mu_{y,i} \sim U(0, 1)$ and $\sigma_i \sim U(0.025, 0.1)$, with 500 samples each for validation and testing.

As shown in Table 3 and Fig. 3, GLNO achieves a relative error of **0.0044**, outperforming Geo-FNO, which serves as an FNO variant for unstructured 2D domains. This superior performance on unstructured meshes validates the mathematical foundation of our LBO-based construction, demonstrating its effectiveness for operator learning and bridging the gap between regular and irregular domains. Besides, GLNO achieves particular improvement in regions exhibiting transient responses, as highlighted in the ground truth visualization.

Table 3: Performance on Geometric Surface. **Bold** indicates the best result. Poisson Equation and Shape-Net are evaluated using $L^2$ error, while other scenarios use ACC for evaluation. "w/o $\sigma$" denotes the ablated variant of our model without the non-periodic basis components.

| Model | Poisson | Shape-Net Car | | SHREC-11 | RNA | Human |
| --- | --- | --- | --- | --- | --- | --- |
| | | Pressure | Velocity | | | |
| MLP (Rumelhart et al., 1986) | 0.4804 | 0.2790 | 0.1293 | 10.3% | 24.7% | 52.9% |
| U-Net (Ronneberger et al., 2015) | 0.2699 | 0.1436 | 0.1267 | 23.3% | 26.1% | 48.3% |
| DiffusionNet (Sharp et al., 2022) | - | - | - | 99.4% | 85.6% | 90.3% |
| Geo-FNO (Li et al., 2023a) | 0.0049 | 0.1278 | 0.1213 | 36.9% | 64.5% | 52.8% |
| GINO (Li et al., 2023b) | 0.1623 | 0.7360 | 0.2563 | 58.3% | 53.9% | 64.1% |
| LSM (Wu et al., 2023) | 0.2612 | 0.7366 | 0.2582 | 42.3% | 33.4% | 57.9% |
| GKNO (Anandkumar et al., 2020) | 0.2486 | 0.1043 | 0.1064 | 55.8% | 51.2% | 38.3% |
| Sp2GNO (Sarkar & Chakraborty, 2025b) | 0.0069 | 0.1197 | 0.1056 | 63.8% | 73.9% | 70.2% |
| GNOT (Hao et al., 2023) | 0.4403 | 0.1109 | 0.1206 | 35.3% | 84.3% | 66.2% |
| Transolver (Wu et al., 2024) | 0.0174 | 0.1098 | 0.1210 | 43.8% | 86.0% | 61.9% |
| AMG (Li et al., 2025) | 0.0152 | 0.0978 | **0.0919** | 55.7% | 88.5% | 62.5% |
| FNO (Li et al., 2020a) | 0.0386 | 0.1272 | 0.1473 | 98.1% | 75.3% | 87.7% |
| GLNO (Ours) | **0.0044** | **0.0960** | 0.1037 | **99.7%** | **90.1%** | **91.0%** |
| GLNO w/o $\sigma$ | 0.0087 | 0.1056 | 0.1452 | 95.5% | 82.2% | 88.5% |

## 5.2 REAL-WORLD DATA MODELING

**Shape Classification on SHREC-11.** We evaluate GLNO on the 30-class SHREC-11 dataset (Lian et al., 2011) with only 10 samples per class to assess its capability in learning global shape structures. This task requires the model to capture intrinsic geometric signatures for classification. GLNO achieves **99.7**% accuracy, significantly outperforming other neural operator methods which show poor performance in this classification setting. This performance gap highlights the necessity of incorporating intrinsic geometric surface information, which our LBO-based basis naturally provides. Moreover, the results demonstrate that GLNO's geometric foundations enable effective learning beyond traditional operator mapping tasks, extending to shape analysis applications.

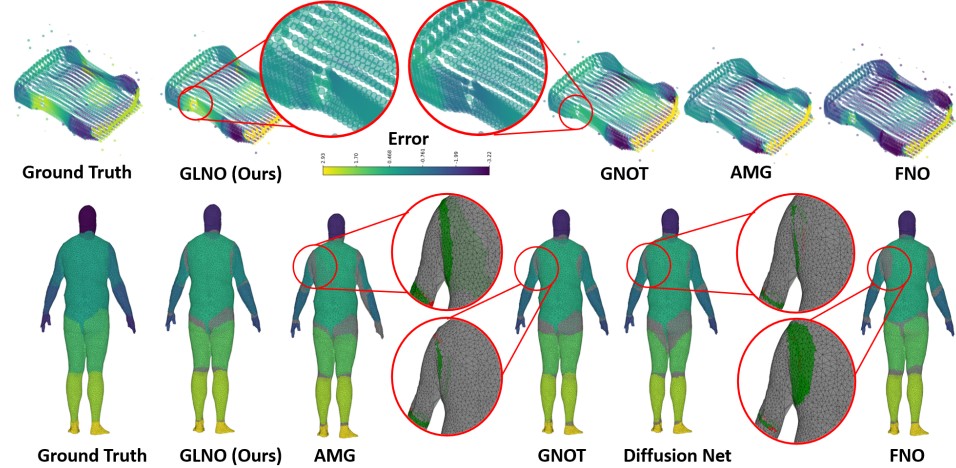

Figure 4: Example results on Shape-Net Car and human body segmentation. Local results of Shape-Net Car is displayed in prediction Error. For local results of human body segmentation, green areas: GLNO outperforms benchmark model; red areas: GLNO underperforms benchmark model.

**Human Body Segmentation and RNA Surface Segmentation.** We utilize the dataset from (Sharp et al., 2022; Maron et al., 2017) featuring 12k-vertex meshes with rotation-augmented coordinates to evaluate surface classification performance. For RNA surface segmentation, we use the dataset from (Poulenard et al., 2019) consisting of 640 RNA surface meshes with approximately 15k vertices, labeled according to 259 atomic categories.

Notably, the training and test sets contain meshes with different vertex counts and significantly different resolutions and topologies, testing the method's generalization across discrete representa-

tions. GLNO achieves $\mathbf{91.0}\%$ accuracy in Human Segmentation and $\mathbf{90.1}\%$ in RNA Segmentation, outperforming both attention-based and graph-based methods.

The superior performance of GLNO is particularly evident in high-curvature regions, as visualized in Fig. 4. This can be attributed to our non-periodic basis based on intrinsic surface properties $\mathcal{P}(x)$, enabling better complex geometric modeling. The results validate the effectiveness of Laplace transform mapping our geometric basis to the complex spectral plane, which enables generalization across different surfaces while maintaining grid invariance.

**Shape-Net Car Simulation.** This task involves predicting surface velocity and pressure for car geometries, simulated at 72 km/h with 889 samples (Li et al., 2025). We learn the mapping from fluid quantities at the last timestep to predicted quantities at the next timestep. GLNO achieves the best performance in pressure prediction ($\mathcal{L}^2 = 0.0960$), while its velocity prediction ($\mathcal{L}^2 = 0.1037$) is slightly behind AMG ($\mathcal{L}^2 = 0.0919$).

This performance pattern can be explained by the relatively limited shape variation in the car dataset compared to the human body and RNA surfaces. The more constrained geometric diversity allows specialized methods like AMG to excel in specific prediction tasks. However, GLNO's overall competitive performance, combined with its demonstrated strength in handling diverse mesh geometries, confirms its robustness as a general-purpose geometric operator.

Table 4: Parameters and training time on Shape-Net Car and human body segmentation datasets.

| Dataset | | Geo-FNO | GINO | LSM | GKNO | Sp2GNO | GNOT | Transolver | AMG | GLNO | GLNO w/o $\sigma$ |
|---|---|---|---|---|---|---|---|---|---|---|---|
| **ShapeNet Car** | Params (M) | 2.37 | 4.74 | 21.68 | 0.29 | 0.65 | 1.39 | 2.26 | 3.54 | 0.21 | 0.21 |
| | Time (s/epoch) | 2.5 | 220 | 18 | 31 | 25 | 166 | 26 | 480 | 2.6 | 2.3 |
| **Human Segmentation** | Params (M) | 9.46 | 4.74 | 21.68 | 0.29 | 0.17 | 0.44 | 2.26 | 3.54 | 0.13 | 0.13 |
| | Time (s/epoch) | 11 | 560 | 19 | 12 | 4.3 | 16 | 13 | 480 | 8.0 | 7.6 |

**Computational Efficiency.** As evidenced in Table 4, GLNO achieves superior performance while maintaining exceptional parameter efficiency (0.13M-0.21M parameters) compared to contemporary methods. The training times are competitive, when considering the method's performance advantages. This efficiency stems from our principled spectral approach, which avoids the expensive graph convolutions or attention mechanisms employed by other geometric learning methods while maintaining strong expressive power.

**Ablation Analysis.** Based on the ablation study where non-periodic bases are removed (GLNO w/o $\sigma$ shown in Table 3), model performance declines across tasks, confirming the importance of non-periodic components in capturing sharp, localized features—especially in high-curvature regions, as visually supported in Fig. 4. The competitive performance of the ablated model in most tasks indicates that the Laplace–Beltrami eigenbasis alone already offers an efficient foundation for representing global behavior. These results consistently verify that integrating global spectral bases with localized non-periodic components yields a more versatile and powerful architecture for operator learning on geometric domains.

## 6 CONCLUSION

We present a novel geometric learning operator by developing a pole-residue framework on the arbitrary Riemannian manifolds. We first propose a generalized operator learning framework based on a pole-residue decomposition enriched with exponential basis functions, enabling expressive modeling of aperiodic and decaying dynamics. Building on this foundation, we introduce the Geometric Laplace Neural Operator (GLNO), which embeds the laplace spectral representation into the eigen-basis of the Laplace–Beltrami operator, extending operator to arbitrary Riemannian manifolds without requiring periodicity or uniform grids. We further design a grid-invariant architecture that realizes GLNO in practice. Extensive experiments on geometric PDEs/ODEs and real-world data modeling demonstrate that GLNO achieves higher accuracy, better generalization, and improved extrapolation performance compared to other state-of-the-art geometric models.

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

APPENDIX

# A    EXPERIMENTAL DETAILS

## A.1    LOSS FUNCTIONS

For regression/prediction tasks, the Relative $\mathcal{L}_2$ Error is used:

$$\text{Relative } \mathcal{L}_2(\mathbf{g}, \mathcal{G}_\theta(f)) = \frac{\|\mathbf{g} - \mathcal{G}_\theta(f)\|_2}{\|\mathbf{g}\|_2} = \sqrt{\frac{\int_{\mathcal{M}} |\mathbf{g}(x) - \mathcal{G}_\theta(f(x))|^2 \mathrm{d}\mu(x)}{\int_{\mathcal{M}} \mathbf{g}(x)^2 \mathrm{d}\mu(x)}}. \tag{21}$$

where $(\mathcal{M}, \mu)$ is the geometry with Riemannian volume element $\mathrm{d}\mu$, $\mathcal{G}_\theta$ is the model with parameters $\theta$ and $f$ is input function and $g$ is target function.

For classification tasks, we use Negative Log-Likelihood Loss (NLL Loss), defined as:

$$\mathcal{L}_{NLL}(\mathbf{g}, \mathcal{G}_\theta(f)) = -\log\left(\frac{\exp(\mathcal{G}_\theta(f)_{\mathbf{g}(x)})}{\sum_{j=1}^{C} \exp(\mathcal{G}_\theta(f)_j)}\right) = \mathcal{G}_\theta(f)_{\mathbf{g}(x)} - \log\left(\sum_{j=1}^{C} \exp(\mathcal{G}_\theta(f)_j)\right) \tag{22}$$

where $\mathbf{g}(x)$ is the labels for each element and $\mathcal{G}_\theta(f)_j$ is the prediction for each label $j \in [1, C]$.

## A.2    DATASET INFORMATION

For the 1D structured grid experiments in Section 5.1, we consider three systems: Driven Pendulum, Duffing Oscillator and Lorenz System. The governing equations for each system are illustrated in Table 2. The dataset consists of 200 training forcing functions $f_{\text{train}}(t) = A\sin(\omega t)$ with $A \in [0.05, 10]$ and 130 test functions $f_{\text{test}}(t) = Ae^{-0.05t}\sin(\omega t)$ with $A \in [0.14, 9.09]$, numerically integrated using the **ode45** solver.

For the 2D structured grid experiments involving the beam equation, diffusion equation, and reaction-diffusion system, we solve each PDE using task-specific forcing functions. The training dataset comprises 200 distinct forcing functions of the form $f_{\text{train}}(x, t) = Ae^{-0.05t}(1 - \omega^2)\sin(\omega x)$, with an additional quadratic term $A^2 e^{-0.1t}\sin^2(\omega x)$ included for the reaction-diffusion task. The amplitude parameter $A$ varies in the range $[0.05, 10]$. Similarly, the test set contains 130 different forcing functions following the same formulation but with modified temporal decay rates: $f_{\text{test}}(x, t) = Ae^{-t}(1 - \omega^2)\sin(\omega x)$ plus the quadratic term $A^2 e^{-2t}\sin^2(\omega x)$ for reaction-diffusion, with $A \in [0.14, 9.09]$.

## A.3    MODEL INFORMATION

Table 5: Model Hyperparameters of GLNO on each real-world experiment

| Parameter | Poisson | Car | SHREC-11 | RNA | Human Body |
|---|---|---|---|---|---|
| Blocks | 4 | 4 | 4 | 4 | 4 |
| Channels | 64 | 64 | 64 | 64 | 64 |
| Poles | 1 | 1 | 4 | 1 | 1 |
| Sigma | 1 | 4 | 4 | 1 | 1 |
| Learning Rate | 1e-3 | 5e-4 | 5e-4 | 1e-3 | 1e-3 |
| Epoch | 300 | 300 | 200 | 300 | 300 |

# B    THEORY ANALYSIS

This section follow the notations in Section 4.

## B.1    KERNEL PARAMETERIZATION IN LAPLACE DOMAIN

The foundation of LNO begins with the kernel integral operator formulation. Consider a general integral operator acting on an input function $f(t)$:

$$(\mathcal{K} * f)(t) = \int_D \kappa(t, x) f(x) \mathrm{d}x \tag{23}$$

**GLNO (Ours)**          **Diffusion Net**

Figure 5: Result comparison on RNA classification task. Different colors represent different labels. Green points in the middle figure represent cases where our model outperforms the Diffusion Net.

Under the assumption of translation invariance, the kernel satisfies $\kappa(t, x) = \kappa(t - x)$, reducing the operator to a convolution:

$$(\mathcal{K} * f)(t) = \int_D \kappa(t - x) f(x) \mathrm{d}x \tag{24}$$

Applying the Laplace transform to both sides and employing the convolution theorem yields:

$$\mathcal{L}\{(\kappa * v)(t)\} = K(s)V(s) \tag{25}$$

where $K(s) = \mathcal{L}\{\kappa(t)\}$ and $V(s) = \mathcal{L}\{v(t)\}$.

### B.2 POLE-RESIDUE COMPUTATION

The multiplication of two Laplace-domain functions in pole-residue form can be decomposed through partial fraction expansion. Consider the product:

$$G(s) = K_\theta(s) \cdot F(s) = \left( \sum_{n=1}^N \frac{\beta_n}{s - \mu_n} \right) \left( \sum_\omega \frac{\alpha_\omega}{s - i\omega} \right) \tag{26}$$

The decomposition into simpler fractions follows from the observation that $G(s)$ is a meromorphic function with poles at $\{\mu_n\} \cup \{i\omega\}$. The general form of the partial fraction expansion is:

$$G(s) = \sum_\omega \frac{\hat{a}_\omega^{\text{steady}}}{s - i\omega} + \sum_{n=1}^N \frac{\hat{a}_n^{\text{transient}}}{s - \mu_n} \tag{27}$$

It can also be interpreted if we apply partial fraction decomposition to separate the contributions from input poles $i\omega$ and kernel poles $\mu_n$:

$$\frac{1}{(s - i\omega)(s - \mu_n)} = \frac{1}{\mu_n - i\omega} \left( \frac{1}{s - \mu_n} - \frac{1}{s - i\omega} \right), \quad \mu_n \neq i\omega \tag{28}$$

The coefficients $\hat{a}_\omega^{\text{steady}}$ and $\hat{a}_n^{\text{transient}}$ are precisely the residues of $G(s)$ at the respective poles. This limit-based approach is fundamental because it isolates the singular part of the function at each pole, capturing the coefficient of the $\frac{1}{s-p}$ term in the Laurent series expansion around each pole $p$.

For poles $\mu_n$:

$$\hat{a}_n^{\text{transient}} = \lim_{s \to \mu_n} (s - \mu_n) G(s) \tag{29}$$

$$= \lim_{s \to \mu_n} (s - \mu_n) \left( \sum_{m=1}^N \frac{\beta_m}{s - \mu_m} \right) \left( \sum_\omega \frac{\alpha_\omega}{s - i\omega} \right) \tag{30}$$

$$= \beta_n \cdot \lim_{s \to \mu_n} \left( \sum_\omega \frac{\alpha_\omega}{s - i\omega} \right) \tag{31}$$

$$= \beta_n \sum_\omega \frac{\alpha_\omega}{\mu_n - i\omega} = \beta_n F(\mu_n) \tag{32}$$

Similarly, for poles $i\omega$:

$$\hat{a}_\omega^{\text{steady}} = \lim_{s \to i\omega} (s - i\omega)G(s) \tag{33}$$

$$= \lim_{s \to i\omega} (s - i\omega)\left(\sum_{n=1}^{N} \frac{\beta_n}{s - \mu_n}\right)\left(\sum_{\omega'} \frac{\alpha_{\omega'}}{s - i\omega'}\right) \tag{34}$$

$$= \alpha_\omega \cdot \lim_{s \to i\omega} \left(\sum_{n=1}^{N} \frac{\beta_n}{s - \mu_n}\right) \tag{35}$$

$$= \alpha_\omega \sum_{n=1}^{N} \frac{\beta_n}{i\omega - \mu_n} = \alpha_\omega K_\theta(i\omega) \tag{36}$$

### B.3 MATHEMATICAL FORMULATION OF GENERALIZED LNO

This appendix provides the complete mathematical derivation of the Laplace Neural Operator (LNO) on Euclidean domains, complementing the generalized Laplace basis framework presented in Section 4.1.

The learnable kernel $K_\theta$ is parameterized in the Laplace spectral domain using a pole-residue formulation:

$$K_\theta(s) = \sum_{n=1}^{N} \frac{\beta_n}{s - \mu_n}, \quad \mu_n \in \mathbb{C}, \beta_n \in \mathbb{C} \tag{37}$$

where $\{\mu_n\}$ are the learnable poles and $\{\beta_n\}$ are the corresponding residues.

The decomposition of input function $f(t)$ onto the generalized Laplace basis $\{\varepsilon_z(t)\}$:

$$\mathcal{D}\{f(t)\} = \sum_{i=1}^{M} \alpha_i \varepsilon_{z_i}(t), \quad \alpha_i = \langle f, \varepsilon_{z_i} \rangle \tag{38}$$

The operator action is defined through multiplication in the spectral domain. Applying the Laplace transform to both the input and the kernel:

$$\mathcal{L}\{\mathcal{D}\{f(t)\}\}(s) = F(s) = \sum_{i=1}^{M} \frac{\alpha_i}{s + z_i} \tag{39}$$

The output in the spectral domain is obtained by pointwise multiplication:

$$G(s) = F(s)K_\theta(s) = \left(\sum_{i=1}^{M} \frac{\alpha_i}{s + z_i}\right)\left(\sum_{n=1}^{N} \frac{\beta_n}{s - \mu_n}\right) \tag{40}$$

Using residue calculus, the output can be decomposed as:

$$G(s) = \sum_{i=1}^{M} \frac{\hat{a}_i^{\text{steady}}}{s + z_i} + \sum_{n=1}^{N} \frac{\hat{a}_n^{\text{transient}}}{s - \mu_n} \tag{41}$$

where the residues are computed analytically:

$$\hat{a}_n^{\text{transient}} = \beta_n F(\mu_n) = \beta_n \sum_{i=1}^{M} \frac{\alpha_i}{\mu_n + z_i} \tag{42}$$

$$\hat{a}_i^{\text{steady}} = \alpha_i K_\theta(-z_i) = \alpha_i \sum_{n=1}^{N} \frac{\beta_n}{-z_i - \mu_n} \tag{43}$$

The final output $g(t)$ is obtained by applying the inverse Laplace transform to $G(s)$. Due to the pole-residue form, this has an analytic solution:

$$g(t) = \mathcal{L}^{-1}\{G(s)\} = \sum_{n=1}^{N} \hat{a}_n^{\text{transient}} e^{\mu_n t} + \sum_{i=1}^{M} \hat{a}_i^{\text{steady}} e^{-z_i t} \tag{44}$$

### B.4 THEOREM: THE LAPLACE TRANSFORM ON GEOMETRY

The mapping $\mathcal{L}\{\varepsilon_z(x)\} \mapsto \frac{1}{s+z}$ can be justified through a rigorous surface integral formulation that extends the classical Laplace transform to Riemannian manifolds.

**Theorem:** Let $(\mathcal{M}, g)$ be a compact Riemannian manifold with volume element $d\mu_g$, and let $\varepsilon_z(x) = e^{-z\mathcal{P}(x)}$ be our generalized basis function, where $\mathcal{P} : \mathcal{M} \to \mathbb{R}$ is a proper function satisfying certain growth conditions. Then the geometric Laplace transform defined by the surface integral:

$$\mathcal{L}\{\varepsilon_z\}(s) = \int_{\mathcal{M}} \varepsilon_z(x) e^{-s\mathcal{P}(x)} d\mu_g(x) \tag{45}$$

satisfies the mapping $\mathcal{L}\{\varepsilon_z\}(s) \mapsto \frac{1}{s+z}$ under appropriate conditions.

**Proof Sketch:**

1. **Parameterization and Jacobian:** Consider a coordinate chart $(U, \varphi)$ on $\mathcal{M}$ such that $\mathcal{P}$ serves as a radial coordinate. The volume element decomposes as $d\mu_g = J(\mathcal{P}, \theta)d\mathcal{P}d\theta$, where $\theta$ represents angular coordinates on level sets of $\mathcal{P}$.

2. **Integral Reformulation:** The geometric Laplace transform becomes:

$$\mathcal{L}\{\varepsilon_z\}(s) = \int_{\mathcal{P}_{\min}}^{\mathcal{P}_{\max}} e^{-(s+z)\mathcal{P}} \left[ \int_{\mathcal{P}^{-1}(\mathcal{P})} J(\mathcal{P}, \theta)d\theta \right] d\mathcal{P} \tag{46}$$

3. **Exponential Decay Condition:** For the integral to converge, we require that the inner integral (the "area" of level sets) grows at most sub-exponentially:

$$A(\mathcal{P}) = \int_{\mathcal{P}^{-1}(\mathcal{P})} J(\mathcal{P}, \theta)d\theta \leq Ce^{k\mathcal{P}} \quad \text{with } k < \text{Re}(s+z) \tag{47}$$

4. **Laplace Transform Analogy:** Under this condition, the transform reduces to a classical Laplace transform of $A(\mathcal{P})$:

$$\mathcal{L}\{\varepsilon_z\}(s) = \mathcal{L}\{A(\mathcal{P})\}(s+z) \tag{48}$$

5. **Constant Area Case:** When $A(\mathcal{P}) \equiv 1$ (corresponding to a flat geometry in the $\mathcal{P}$-coordinate), we recover the exact mapping:

$$\mathcal{L}\{\varepsilon_z\}(s) = \int_0^{\infty} e^{-(s+z)\mathcal{P}}d\mathcal{P} = \frac{1}{s+z} \tag{49}$$

6. **Perturbation Analysis:** For general manifolds where $A(\mathcal{P})$ deviates from constant, the mapping holds approximately when the variations in $A(\mathcal{P})$ are small compared to the exponential decay rate $\text{Re}(s+z)$.

This surface integral formulation provides the mathematical foundation for our geometric Laplace transform, showing that the mapping $\varepsilon_z(x) \mapsto \frac{1}{s+z}$ emerges naturally from the geometric structure when $\mathcal{P}$ serves as an appropriate "time-like" coordinate on the manifold.

**Theorem:** Under the assumptions of manifold homogeneity and radial symmetry of $\kappa$, the geometric Laplace transform maps convolution to pointwise multiplication:

$$\mathcal{L}\{\kappa * f\}(s) = K(s) \cdot \mathcal{L}\{f\}(s) \tag{50}$$

where $K(s)$ is the geometric Laplace transform of the kernel $\kappa$.

**Proof:**

We begin with the convolution operator on the homogeneous manifold:

$$(\kappa * f)(x) = \int_{\mathcal{M}} \kappa(d_g(x, y))f(y)d\mu_g(y) \tag{51}$$

Using the coordinate system where $d\mu_g = A(\mathcal{P})d\mathcal{P}d\theta$ and the translation invariance $\kappa(d_g(x,y)) = \kappa(|\mathcal{P}(x) - \mathcal{P}(y)|)$, we rewrite the convolution as:

$$(\kappa * f)(x) = \int_0^\infty \kappa(|\mathcal{P}(x) - \mathcal{P}|) \left[ \int_{\mathcal{P}^{-1}(\mathcal{P})} f(y)A(\mathcal{P})d\theta \right] d\mathcal{P} \tag{52}$$

Define the radial projection of $f$ as:

$$F(\mathcal{P}) = \int_{\mathcal{P}^{-1}(\mathcal{P})} f(y)A(\mathcal{P})d\theta \tag{53}$$

Then the convolution simplifies to a one-dimensional form:

$$(\kappa * f)(\mathcal{P}(x)) = \int_0^\infty \kappa(|\mathcal{P}(x) - \mathcal{P}'|)F(\mathcal{P}')d\mathcal{P}' \tag{54}$$

Now apply the geometric Laplace transform to both sides. For the left-hand side:

$$\mathcal{L}\{\kappa * f\}(s) = \int_0^\infty (\kappa * f)(\mathcal{P}(x))e^{-s\mathcal{P}(x)}A(\mathcal{P}(x))d\mathcal{P}(x) \tag{55}$$

Substitute the convolution expression:

$$\mathcal{L}\{\kappa * f\}(s) = \int_0^\infty \left[ \int_0^\infty \kappa(|\mathcal{P}(x) - \mathcal{P}'|)F(\mathcal{P}')d\mathcal{P}' \right] e^{-s\mathcal{P}(x)}A(\mathcal{P}(x))d\mathcal{P}(x) \tag{56}$$

$$= \int_0^\infty F(\mathcal{P}') \left[ \int_0^\infty \kappa(|\mathcal{P}(x) - \mathcal{P}'|)e^{-s\mathcal{P}(x)}A(\mathcal{P}(x))d\mathcal{P}(x) \right] d\mathcal{P}' \tag{57}$$

Make the substitution $u = \mathcal{P}(x) - \mathcal{P}'$ in the inner integral:

$$\int_0^\infty \kappa(|\mathcal{P}(x) - \mathcal{P}'|)e^{-s\mathcal{P}(x)}A(\mathcal{P}(x))d\mathcal{P}(x) \tag{58}$$

$$= e^{-s\mathcal{P}'} \int_{-\mathcal{P}'}^\infty \kappa(|u|)e^{-su}A(u + \mathcal{P}')du \tag{59}$$

Under the assumption that $A(\mathcal{P})$ varies slowly compared to the exponential decay (or that $A(\mathcal{P}) \approx 1$ for the region where $\kappa(u)$ is significant), we approximate $A(u + \mathcal{P}') \approx A(\mathcal{P}')$. Then:

$$\approx e^{-s\mathcal{P}'}A(\mathcal{P}') \int_{-\infty}^\infty \kappa(|u|)e^{-su}du \tag{60}$$

$$= e^{-s\mathcal{P}'}A(\mathcal{P}')K(s) \tag{61}$$

where $K(s) = \int_{-\infty}^\infty \kappa(|u|)e^{-su}du$ is the Laplace transform of the radial kernel.

Substituting back:

$$\mathcal{L}\{\kappa * f\}(s) = \int_0^\infty F(\mathcal{P}')e^{-s\mathcal{P}'}A(\mathcal{P}')K(s)d\mathcal{P}' \tag{62}$$

$$= K(s) \int_0^\infty F(\mathcal{P}')e^{-s\mathcal{P}'}A(\mathcal{P}')d\mathcal{P}' \tag{63}$$

From the definition of $F(\mathcal{P}')$ and the geometric Laplace transform:

$$\int_0^\infty F(\mathcal{P}')e^{-s\mathcal{P}'}A(\mathcal{P}')d\mathcal{P}' = \mathcal{L}\{f\}(s) \tag{64}$$

Therefore, we obtain the desired multiplicative result:

$$\mathcal{L}\{\kappa * f\}(s) = K(s) \cdot \mathcal{L}\{f\}(s) \tag{65}$$

In the special case where $A(\mathcal{P}) \equiv 1$ (flat geometry), the approximation becomes exact and we recover the classical Laplace convolution theorem.

## B.5 COMPUTATIONAL COMPLEXITY ANALYSIS

We analyze the forward-pass complexity of the proposed GLNO in the one-dimensional temporal setting, which is similar for higher-dimensional settings and meshes.

**Notation:** We summarize the main symbols used in this subsection:

- $K$: number of discretization points in the temporal domain $[0, T]$;
- $M$: number of generalized Laplace basis functions $\{\varepsilon_{z_i}\}_{i=1}^{M}$;
- $N$: number of poles in the learnable kernel $K_\theta(s) = \sum_{n=1}^{N} \frac{\beta_n}{s-\mu_n}$;
- $D$: channel dimension;

**Generalized Laplace Decomposition** For each input signal $f(t) \in \mathbb{R}^{d_{\text{in}}}$ sampled on $K$ grid points, we compute the generalized Laplace coefficients in Equation 8, implemented by multiplying $f(t)$ with $e^{-\sigma_i t}$ and then applying an FFT with respect to the oscillatory part $e^{-i\omega_i t}$.

A implementation uses one FFT per $\sigma_i$, leading to a per-sample cost

$$\text{Cost}_{\text{decomp}} = \mathcal{O}\big(MK \log K \cdot D\big)$$

**Laplace-Domain Kernel Multiplication** The dominant cost in Equation 15 comes from forming all $M \times N$ interactions between input basis coefficients $\{\alpha_i\}$ and kernel poles $\{\mu_n, \beta_n\}$, which scales as

$$\text{Cost}_{\text{kernel}} = \mathcal{O}\big(MN \cdot D\big)$$

**Inverse Laplace Transform and Reconstruction** After spectral multiplication, the output in the time domain is reconstructed as

$$g_\theta(t_k) = \sum_{i=1}^{M} \hat{a}_i^{\text{steady}} e^{-z_i t_k} + \sum_{n=1}^{N} \hat{a}_n^{\text{transient}} e^{\mu_n t_k}, \qquad k = 1, \dots, K$$

where $\hat{a}_i^{\text{steady}}$ and $\hat{a}_n^{\text{transient}}$ are the residues determined by $(\alpha_i, \beta_n, \mu_n, z_i)$. Evaluating this expression on all $K$ time steps and $d_{\text{out}}$ output channels costs

$$\text{Cost}_{\text{recon}} = \mathcal{O}\big((M + N)K \cdot D\big)$$

per sample.

**Total Forward Complexity** Combining all three stages, the overall forward-pass complexity satisfies

$$\text{C}_{\text{forward}} = \mathcal{O}\Big(D(MK \log K + MN + (M + N)K)\Big) \approx \mathcal{O}(DMK \log K)$$

for many practical regimes where $K$ is large while $M$ and $N$ are moderate.

## B.6 APPROXIMATION BOUNDS WITH FINITE PARAMETERS

We consider the one-dimensional temporal setting $t \in [0, T]$ and study the approximation properties when applied to linear time-invariant operators.

**Target operator** Let $\mathcal{T}_* : L^2([0, T]) \to L^2([0, T])$ be a bounded linear operator with convolution kernel $k_* \in L^2([0, T])$:

$$(\mathcal{T}_* f)(t) = \int_0^T k_*(t, \tau) f(\tau)\, \mathrm{d}\tau$$

Let $H_*(s)$ denote the Laplace transform of $k_*$,

$$H_*(s) = \mathcal{L}\{k_*\}(s)$$

which is not assumed to be rational (i.e., $H_*$ may have infinitely many poles or an essential singularity).

**Notation:** Following the output in Equation 11, we summarize the main symbols used in this subsection:

- $M$: number of generalized Laplace basis functions $\{\varepsilon_{z_i}\}_{i=1}^M$;

- $N$: number of poles in the learnable kernel $K_\theta(s) = \sum_{n=1}^N \frac{\beta_n}{s - \mu_n}$;

- $c_i(\theta), d_n(\theta) \in \mathbb{C}$ are coefficient functions determined by learnable parameters $\theta \in \mathbb{R}^{d_\theta}$.

The corresponding learned operator is

$$(\mathcal{T}_\theta f)(t) = \int_0^T k_\theta(t - \tau) f(\tau) \, \mathrm{d}\tau$$

**Approximation errors** We decompose the total error into two parts:

$$\|\mathcal{T}_* - \mathcal{T}_\theta\|_{L^2 \to L^2} \leq \varepsilon_{\mathrm{basis}}(M, N) + \varepsilon_{\mathrm{par}}(d_\theta)$$

where:

- $\varepsilon_{\mathrm{basis}}(M, N)$ is the best approximation error of $k_*$ by finite exponential combinations,

$$\varepsilon_{\mathrm{basis}}(M, N) = \inf_{\{c_i, d_n, z_i, \mu_n\}} \left\| k_* - \sum_{i=1}^M c_i e^{-z_i t} - \sum_{n=1}^N d_n e^{\mu_n t} \right\|_{L^2(0,T)}$$

- $\varepsilon_{\mathrm{par}}(d_\theta)$ is the finite-parameterization error,

$$\varepsilon_{\mathrm{par}}(d_\theta) = \inf_{\theta \in \mathbb{R}^{d_\theta}} \left\| k_{M,N}^{\mathrm{opt}} - k_\theta \right\|_{L^2(0,T)}$$

where $k_{M,N}^{\mathrm{opt}}$ denotes any minimizer in the definition of $\varepsilon_{\mathrm{basis}}(M, N)$.

**Non-rational kernel approximation** Assume $k_* \in H^\alpha(0, T)$ for some $\alpha > 0$, i.e., the true kernel has Sobolev regularity $\alpha$. Classical results on exponential approximations imply the existence of coefficients and exponents such that

$$\varepsilon_{\mathrm{basis}}(M, N) \leq C_1 (M + N)^{-\alpha}$$

for some constant $C_1$ independent of $M, N$. Equivalently, even if $H_*(s)$ is not rational, its time-domain kernel $k_*$ can be approximated in $L^2(0, T)$ by a finite sum of generalized exponentials at a Sobolev-type rate depending on $\alpha$.

By Young's inequality for convolutions, the induced operator error satisfies

$$\|\mathcal{T}_* - \mathcal{T}_{M,N}^{\mathrm{opt}}\|_{L^2 \to L^2} \leq \|k_* - k_{M,N}^{\mathrm{opt}}\|_{L^1(0,T)} \leq C' \varepsilon_{\mathrm{basis}}(M, N)$$

**Finite-parameterization error** Assume the map $\theta \mapsto k_\theta$ is $L$-Lipschitz from $(\mathbb{R}^{d_\theta}, \|\cdot\|_2)$ to $(L^2(0, T), \|\cdot\|_{L^2})$. Then standard finite-dimensional approximation theory for neural networks (or any parametric family) yields

$$\varepsilon_{\mathrm{par}}(d_\theta) \leq C_2 \, d_\theta^{-p}$$

where $p > 0$ depends on the architecture (depth, activation, and the regularity of $k_{M,N}^{\mathrm{opt}}$ as a function of the parameters).

**Final approximation** Combining the above estimates, we obtain the following bound for GLNO:

$$\|\mathcal{T}_* - \mathcal{T}_\theta\|_{L^2 \to L^2} \leq C\left((M + N)^{-\alpha} + d_\theta^{-p}\right)$$

for some constant $C > 0$ independent of $M, N$, and $d_\theta$. In particular, the generalized Laplace basis provides a controllable trade-off between the number of temporal modes $(M + N)$ and parameter count $d_\theta$, even when the true Laplace-domain kernel $H_*(s)$ is not rational.

### B.7 SAMPLE COMPLEXITY

The class of all GLNO operators with $M$ basis modes and parameter dimension $d_\theta$ is:

$$\mathcal{H}_{\text{GLNO}} = \left\{ \mathcal{G}_\theta : \ \mathcal{G}_\theta(u) = \sum_{k=1}^{M} a_k(\theta) \left( \phi_k * u \right) \right\} \tag{66}$$

where $\{\phi_k\}_{k=1}^{M}$ are basis functions and $a_k(\theta)$ are neural coefficients parameterized by a $d_\theta$-dimensional network. Given data samples

$$\{(u_i, \ \mathcal{G}_*(u_i))\}_{i=1}^{N}$$

the empirical risk is

$$\widehat{\mathcal{R}}(\theta) = \frac{1}{N} \sum_{i=1}^{N} \|\mathcal{G}_\theta(u_i) - \mathcal{G}_*(u_i)\|_{L^2(\Omega)}^2 \tag{67}$$

**Rademacher Complexity**   Since the mapping $u \mapsto (\phi_k * u)$ is linear and bounded, and the coefficient vector $a(\theta)$ is parametrized by a $d_\theta$-parameter neural network, the hypothesis class satisfies the complexity bound:

$$\mathfrak{R}_N(\mathcal{H}_{\text{GLNO}}) = \mathcal{O}\left( \sqrt{\frac{M + d_\theta}{N}} \right) \tag{68}$$

This scaling reflects that $M$ controls the number of operator basis modes and $d_\theta$ controls the expressive capacity of the coefficient mapping.

**Generalization Bound**   With probability at least $1 - \delta$, the GLNO satisfies

$$\mathcal{R}(\theta) \leq \widehat{\mathcal{R}}(\theta) + C\sqrt{\frac{M + d_\theta + \log(1/\delta)}{N}} \tag{69}$$

where $\mathcal{R}(\theta)$ denotes the population risk. Thus the generalization behavior depends jointly on the basis size $M$ and the parameter dimension $d_\theta$.

**Minimum Required Samples**   To achieve accuracy $\epsilon$ in population risk, it suffices that

$$\sqrt{\frac{M + d_\theta}{N}} \leq \epsilon$$

which yields the sample requirement:

$$N_{\min}^{\text{GLNO}} = \mathcal{O}\left( \frac{M + d_\theta}{\epsilon^2} \right) \tag{70}$$

Thus the number of required samples grows linearly with the basis size of the operator and the network size of the coefficient mapping, matching classical operator-learning scaling laws.

## C   THE USAGE OF LLMS

In this work, the large language models are only used for writing polishing, helping us check grammar, organize sentence structures and unify word usage in the final draft stage, thus making the paper more readable. The core content is fully led and verified by the author throughout the process.

## D   REPRODUCIBILITY

We will release the full codes from GitHub upon acceptance.

