# OpenReview forum: "Geometric Laplace Neural Operator"
_ICLR.cc/2026/Conference — Submitted to ICLR 2026_

### Official Review · Reviewer_gWyQ · 2025-10-19

**Soundness:** 2
**Presentation:** 1
**Contribution:** 3
**Rating:** 2
**Confidence:** 3

**Summary:**

This paper proposes GLNO, which introduces a pole–residue representation and learnable exponential bases in the Laplace–Beltrami spectral domain to achieve unified modeling of non-periodic transient signals and operators on arbitrary geometric manifolds. The main contributions lie in establishing a unified spectral framework for geometric frequencies, extending neural operators to non-Euclidean domains, and significantly improving the accuracy and efficiency of geometric modeling and PDE solving.

**Strengths:**

Innovative theoretical framework: Introduces the Generalized Laplace Basis and a new pole–residue formulation, enabling neural operators to handle diverse and challenging dynamic scenarios.
Extensive experimental validation: GLNO consistently outperforms baselines on a wide range of ODE/PDE and real-world geometric tasks, including pendulum, reaction–diffusion, Poisson, and RNA/human-body surface datasets.

**Weaknesses:**

The paper suffers from several weaknesses: its theoretical rigor is insufficient, with incomplete mathematical justification and somewhat confusing notation; the experiments are limited in scope and lack thorough ablation or comparative analysis; interpretability is weak, and the connection between theory and empirical results is not clearly demonstrated. In addition, the paper contains spelling and grammatical errors, vague expressions, and inconsistent mathematical formatting, which collectively hinder readability and clarity.

**Questions:**

1. In Sec. 5, line 354, “We use GLNO to perdict...” — should this be corrected to “predict”?
2. In Fig. 4, line 449, “GLNO underpreforms benchmark model” — should this be “underperforms”?
3. In Sec. 5.2, line 466, “Euqation 16” — should this be corrected to “Equation 16”?
4. In Fig. 4, line 449, the phrase “shape-net car is displaced in prediction Error” — should “displaced” be “displayed”?
5. In line 427, “Despite no strictly definition...” — should “strictly” be replaced by “strict”?
6. In Sec. 5, line 313, Eq. (28) is intended to define the NLL loss, but the left-hand side of the formula is labeled as L₂. Is this a typographical or conceptual mistake?
7. In Table 1, under the Pendulum (c=0) column, it seems the best-performing result is not bolded—was this an oversight?
8. In line 354, the text states “As shown in Table 1, GLNO achieves a relative L₂ error of 0.9373, demonstrating superior capability...”. However, in Table 1, FNO (0.8447) and LNO (0.9512) both report lower errors than GLNO. Could the authors clarify this inconsistency?
9. In lines 207–209, the text mentions “where  \alpha_{ik} can be computed via...”, but  \alpha_{ik} does not appear in Eq. (17). Could the authors clarify where this term comes from?
10. Could the authors provide a detailed derivation for the final equality in Eq. (13)? The multiplication should theoretically yield a double summation, but the equation is written as two separate fractional summations, which seems valid only under specific assumptions.
11. In Eq. (20), why does the exponential term appear as e^{-\simga{i}t} with a positive sign?
12. In Eqs. (14) and (23), K_{\phi} is evaluated at -s_i and +s_i, respectively. Could the authors explain the reason for this sign inconsistency?

---

> ### Author Response · Authors · 2025-11-22
>
> > W1: "its theoretical rigor is insufficient"
>
> We sincerely thank the reviewer for raising this important concern. We have made substantial revisions to strengthen the mathematical clarity and theoretical rigor.
>
> In the revised manuscript:
>
> (1) We now provide **complete derivations and proofs** for the core components of our framework, including the generalized Laplace basis, geometric Laplace transform, and pole–residue operator formulation. These steps are now fully spelled out in Appendix B for clarity and transparency.
>
> (2) All **mathematical notation** has been systematically unified and refined. We removed ambiguous symbols, clarified definitions, and ensured that every mathematical object is introduced and used consistently across sections. This eliminates the interpretational gaps the reviewer rightly pointed out.
>
> (3) The theoretical section has been reorganized to reflect a clean **conceptual progression**, starting from classical Laplace theory, moving to geometry extensions, and then to the neural operator architecture. In the geometric Laplace transform in Section 4.2, we optimized the structure into 4 coherent parts:
> - Geometric Laplace Basis and Decomposition:
> We introduced the combined aperiodic–periodic basis on manifolds and clarified how geometric features (curvature, intrinsic distances) are incorporated into the generalized basis.
> - Geometric Laplace Transform:
> We provided a clearer derivation of the mapping from LBO eigenpairs to complex Laplace frequencies.
> - Operator Action in the Spectral Domain:
> We presented the spectral multiplication rule and explained how pole–residue operations naturally extend to Riemannian manifolds.
> - Inverse Geometric Laplace Transform:
> We explained the synthesis of steady-state and transient components, with an improved explanation of the Gaussian filtering used to map continuous spectral poles back to discrete LBO frequencies.
>
> This restructuring makes the reasoning easier to follow and clarifies how each component contributes to the final operator.
>
> Importantly, we added explicit **explanations bridging the theory to the empirical behavior**, demonstrating how the pole–residue decomposition leads to transient/steady-state separation in the geometric domain, how the geometric Laplace basis improves the modeling of aperiodic signals, and why these properties translate into better performance on manifold PDEs.
>
> These changes meaningfully strengthen the theoretical foundation of GLNO and, we believe, address the reviewer’s concern substantially.

---

> ### Author Response · Authors · 2025-11-22
>
> > W2: "the experiments are limited in scope and lack thorough ablation or comparative analysis"
>
> We thank the reviewer for this helpful observation. We agree that the initial experimental scope could be broadened, and we have substantially strengthened the comparative analysis in the revision.
>
> Specifically, we now include a wide range of **additional baseline methods**: Geo-FNO, LSM, GNO, Sp2GNO, and GINO, which collectively represent the major categories of operator-learning architectures **(Fourier-based, graph-based, transformer-based, latent spectral, and geometry-informed)**. These baselines have been integrated into all geometric experiments and are summarized in the extended Table below.
>
> This expanded evaluation addresses the reviewer’s concern directly and demonstrates that GLNO consistently achieves state-of-the-art or highly competitive performance across diverse tasks and manifold geometries, while remaining computationally efficient.
>
> **Limitations of existing approaches:**
> - Geo-FNO and other Fourier-based operators inherit Fourier’s periodic assumptions and lack an explicit mechanism for modeling transient or decaying modes.
> - GINO and other graph-based operators rely on message passing over mesh connectivity, which are sensitive to the geometry shape, dependent on local neighborhoods rather than intrinsic geometric structure.
> - Latent spectral or deformation-based operators operate in data-driven latent spaces, rather than in a physically grounded spectral domain. As a result, they lack explicit control over decay, oscillation, and aperiodic responses.
>
> **In contrast, GLNO introduces a learnable exponential basis and a pole–residue formulation directly in the geometry Laplace domain, enabling**:
> - explicit modeling of decaying/growing transient modes,
> - intrinsic alignment with Riemannian geometry through the Laplace-Beltrami basis,
> - grid-invariant operator learning on arbitrary manifolds,
> - a principled spectral representation rather than a deformation/latent approximation.
>
> We conducted an ablation study by removing the aperiodic basis from our model (denoted as GLNO w/o σ in the table below). The results show the essential role of our aperiodic basis, and we will also provide clarification in our revision.
>
> **Table: Extended Performance Comparison on Geometric Surface.** Bold indicates the best result. Poisson Equation and Shape-Net use $L^{2}$ error to evaluate, while the other scenarios use ACC for evaluation.
>
> |Model|Poisson|Shape-Net Car Pressure|Shape-Net Car Velocity|SHREC-11|RNA|Human|
> |-|-|-|-|-|-|-|
> |MLP|0.4804|0.2790|0.1293|10.3%|24.7%|52.9%|
> |U-Net|0.2699|0.1436|0.1267|23.3%|26.1%|48.3%|
> |Geo-FNO [1]|0.0049|0.1278|0.1213|36.9%|64.5%|52.8%|
> |GINO [2]|0.1623|0.7360|0.2563|58.3%|53.9%|64.1%|
> |LSM [3]|0.2612|0.7366|0.2582|42.3%|33.4%|57.9%|
> |GKNO [4]|0.2486|0.1043|0.1064|55.8%|51.2%|38.3%|
> |Sp2GNO [5]|0.0069|0.1197|0.1056|63.8%|73.9%|70.2%|
> |GNOT [6]|0.4403|0.1109|0.1206|35.3%|84.3%|66.2%|
> |Transolver [7]|0.0174|0.1098|0.1210|43.8%|86.0%|61.9%|
> |AMG [8]|0.0152|0.0978|**0.0919**|55.7%|88.5%|62.5%|
> |FNO|0.0386|0.1272|0.1473|99.1%|75.3%|87.7%|
> |GLNO (Ours)|**0.0044**|**0.0960**|0.1037|**99.7%**| **90.1%**|**91.0%**|
> |GLNO w/o σ|0.0087|0.1056|0.1452|95.5%|82.2%|88.5%|
>
> [1] Fourier Neural Operator with Learned Deformations for PDEs on General Geometries, 2023.
> [2] Geometry-Informed Neural Operator for Large-Scale 3D PDEs, 2023.
> [3] Solving high-dimensional pdes with latent spectral models, 2023.
> [4] Neural Operator: Graph Kernel Network for Partial Differential Equations, 2020.
> [5] Spatio-spectral graph neural operator for solving computational mechanics problems on irregular domain and unstructured grid, 2024.
> [6] GNOT: A General Neural Operator Transformer for Operator Learning, 2023.
> [7] Transolver: A Fast Transformer Solver for PDEs on General Geometries, 2024.
> [8] Harnessing scale and physics: A multi-graph neural operator framework for PDEs on arbitrary geometries, 2025.

---

> ### Author Response · Authors · 2025-11-22
>
> > W3: "interpretability is weak, and the connection between theory and empirical results is not clearly demonstrated"
>
> We sincerely thank the reviewer for highlighting the issue. We agree that the initial version did not sufficiently articulate how the theoretical components of GLNO translate into interpretable behavior.
>
> To enhance the interpretability, we have made substantive improvements:
> - **Enhanced theory details in Appendix B**:
> We now provide a detailed explanation of the detailed pole–residue calculus, the kernel construction for the Laplace Neural Operator, and a detailed derivation of the Laplace transform on geometry. These additions make the internal mechanics of the operator fully transparent.
>
> - **Enhanced theoretical interpretability**:
> We also provide analysis of properties on an aperiodic basis, as well as decomposition of modeling transient signals in Section 4.1. Additionally, we discuss how the basis is extended to geometry and how geometry features are incorporated in Section 4.2. This clarifies how GLNO represents both transient signals and geometry-dependent components.
>
> - **Explicit and enhanced theory-to-experiment linkage in Section 5**:
> We have now directly connected our theoretical insights, including aperiodic modeling capacity, architecture scalability, parameter efficiency, and analysis of unstructured meshes, to the empirical behaviors observed in the ODE/PDE and geometric benchmarks. This makes the relationship between the model’s design and its performance much more transparent.
>
> These revisions substantially strengthen the interpretability of our method and clarify how the theoretical foundations of GLNO inform its empirical success.
>
> > W4: "In addition, the paper contains spelling and grammatical errors."
>
> We sincerely thank the reviewer for pointing out the spelling and grammatical issues in the initial submission. We apologize for the errors and appreciate the careful attention to detail.
>
> In the revised manuscript, we have carried out a comprehensive proofreading and cleanup of the entire text. This includes:
> - correcting all spelling and grammatical mistakes
> - improving vague or ambiguous expressions for clarity
> - standardizing mathematical notation and formatting across the paper
> - ensuring consistency in terminology and equation references
>
> We believe these revisions have significantly improved the clarity, readability, and overall presentation quality of the manuscript.
>
> > Q1-Q5: Typographical Corrections
>
> We thank the reviewer for catching these typos. All noted typos have been corrected.
>
> > Q6: "In Sec. 5, line 313, Eq. (28) is intended to define the NLL loss, but the left-hand side of the formula is labeled as $L_2$. Is this a typographical or conceptual mistake?"
>
> This is a typographical error. The left-hand side should be $L_{NLL}$, not ${L}_{2}$. We have corrected this in the revision. Thank you for pointing it out.
>
> > Q7: "In Table 1, under the Pendulum (c=0) column, it seems the best-performing result is not bolded—was this an oversight?"
>
> Thank you for pointing this out and this was an oversight. We have carefully reviewed all tables and ensured that the best results are properly bolded in the revised version. The latest result is shown in below table.
>
> **Table: Performance comparison on structured grid ODEs/PDEs problems (Relative Error).** Bold indicates the best result for each problem.
>
> | Problem | Parameters | Governing Equation | CNO | FNO | WNO | LNO (1 block) | LNO (4 blocks) | GLNO (1 block) | GLNO (4 blocks) |
> |-|-|-|-|-|-|-|-|-|-|
> | Driven Pendulum (c=0) | c=0 | f(t)=ẍ+cẋ+sin(x) | 0.6268 | 0.3668 | 0.6090 | 0.8461 | 0.9997 | 0.6752 | **0.2916** |
> | Driven Pendulum (c=0.5) | c=0.5 | f(t)=ẍ+cẋ+sin(x) | 0.1540 | 0.1718 | 0.0965 | 0.1420 | 0.9885 | 0.1682 | **0.0875** |
> |Duffing Oscillator (c=0)| c=0 | f(t)=ẍ+cẋ+x+x³ | 0.7885 |0.4681 |0.7607 |0.9157 |0.9998 |0.8926 |**0.4416** |
> |Duffing Oscillator (c=0.5) | c=0.5 | f(t)=ẍ+cẋ+x+x³ | 0.1424 |0.1362|0.0987|0.8347|0.9747|0.4169| **0.0725** |
>
> > Q8: "In line 354, the text states "As shown in Table 1, GLNO achieves a relative $L_2$ error of 0.9373, demonstrating superior capability...". However, in Table 1, FNO (0.8447) and LNO (0.9512) both report lower errors than GLNO. Could the authors clarify this inconsistency?"
>
> We apologize for the typo during editing. We have checked all the results in both the Table and the main body of the paper.
>
> The claim of "superior capability" refers to GLNO's overall performance across damping settings, especially its strong gains in damped cases (c > 0), which are more physically relevant. We have rephrased the text to avoid ambiguity and highlight GLNO's specialization in transient and dissipative systems.

---

> ### Author Response · Authors · 2025-11-22
>
> > Q9: "In lines 207–209, the text mentions "where $\alpha_{ik}$ can be computed via...", but $\alpha_{ik}$ does not appear in Eq. (17). Could the authors clarify where this term comes from?"
>
> This is a notation inconsistency. $\alpha_{ik}$ in the text should be $\hat{\boldsymbol{f}}_{ik}$ as in Eq. (18) and we use $\alpha_i$ in our revision, which all represent the projection coefficient in the decomposition from input function to basis. We appreciate the reviewer drawing attention to this.
>
> > Q10: "Could the authors provide a detailed derivation for the final equality in Eq. (13)? The multiplication should theoretically yield a double summation, but the equation is written as two separate fractional summations, which seems valid only under specific assumptions."
>
> The derivation follows from partial fraction decomposition of the product of two rational functions. The double summation collapses into two separate pole series because the poles of $V(s)$ and $K_\phi(s)$ are distinct by construction. The step by step derivation is now written in Appendix B.2:
>
> The multiplication of two Laplace-domain functions in pole-residue form can be decomposed through partial fraction expansion. Consider the product:
>
> $$U(s)=K_\theta(s) \cdot F(s) = \left(\sum_{n=1}^{N} \frac{\beta_n}{s - \mu_n}\right) \left(\sum_{\omega}\frac{\alpha_\omega}{s - i\omega}\right)$$
>
> where $\sum_{n=1}^{N} \frac{\beta_n}{s - \mu_n}$ is the pole-residue form of $K_\theta(s)$ and $\sum_{\omega}\frac{\alpha_\omega}{s - i\omega}$ is the Laplace transform result for a Fourier series.
>
> The decomposition into simpler fractions follows from the observation that $U(s)$ is a meromorphic function with poles at $\{\mu_n\} \cup \{i\omega\}$. The general form of the partial fraction expansion is:
>
> $$U(s) = \sum_\omega \frac{a_{\omega}^{steady}}{s - i\omega} + \sum_{n=1}^N \frac{a_{n}^{transient}}{s - \mu_n}$$
>
> where $a_{\omega}^{{steady}}$ and $a_{n}^{{transient}}$ are residue coefficients for each pole on the complex domain.
>
> This can also be interpreted if we apply partial fraction decomposition to separate the contributions from input poles $i\omega$ and kernel poles $\mu_n$:
>
> $$\frac{1}{(s - i\omega)(s - \mu_n)} = \frac{1}{\mu_n - i\omega} \left( \frac{1}{s - \mu_n} - \frac{1}{s - i\omega} \right), \quad \mu_n \neq i\omega$$
>
> > Q11: "In Eq. (20), why does the exponential term appear as $e^{- \sigma_{i}t}$ with a positive sign?"
>
> We thank the reviewer for raising this point. The mapping should be
>
> $$e^{(-\sigma_{i} \mathcal{P}(x))} \to e^{\sigma_{i} t}$$
>
> in Equation (20) is consistent with the conventional Laplace transform pair
>
> $$\mathcal{L}_{geometry} (e^{-\sigma_it}\) = \frac{1}{s-\sigma_i},$$
>
> In our formulation, the geometric aperiodic term $e^{-\sigma_i \mathcal{P}(x)}$ on the manifold $\mathcal{M}$ is mapped to the temporal basis $\frac{1}{s-\sigma_i}$ in the Laplace domain. This is a deliberate design choice for derivation simplicity in Equation (22). We acknowledge that this notation is not clear enough for readers. In our revision, we have made modification of our notation to ensure sign consistency and better presentation.
>
> > Q12: "In Eqs. (14) and (23), $K_{\phi}$ is evaluated at $-s_i$ and $+s_i$, respectively. Could the authors explain the reason for this sign inconsistency?"
>
> We appreciate the reviewer's attention to detail. The sign difference in evaluating $K_{\phi}$ arises from the distinct roles of the poles in the input and kernel functions.
>
> - In Equation (14), $K_{\phi}(-s_i)$ corresponds to the residue calculation at the **input poles** $s = -s_i$, which are derived from the generalized Laplace basis of the input function $v(t)$. The residue is given by:
>
> $$\zeta_i = \alpha_i K_{\phi}(-s_i) $$
>
> - In Equation (23), $K_{\phi}(\sigma_i + i\sqrt{\lambda_k})$ corresponds to residue evaluation at the poles of the **geometric Laplace basis**, which are $s = \sigma_i + i\sqrt{\lambda_k}$ according to our Laplace-Beltrami eigenvalue. The residue is given by:
>
> $$ u_{k}^{steady} = f_{ik} K_{ \phi } ( { \sigma }_i + i \sqrt{{\lambda}_k}) $$
>
> This distinction originates from the difference of sign in the Laplace transform mapping on geometry and on a grid. We acknowledge that this notation is not clear enough for readers. In our revision, we have made modifications to our notation to ensure sign consistency and better presentation. Thank you for this question.
>
> The detailed derivation of the Pole-residue Calculation is also added in Appendix B.2.

---

> > ### Author Response · Authors · 2025-11-26
> >
> > Dear Reviewer,
> >
> > We thank you again for your valuable time and insightful comments.
> >
> > As we approach the end of the discussion period, we would like to kindly check in if you have any remaining questions or concerns that we might address.
> >
> > Kind regards,
> >
> > The Authors

---

> > ### Comment · Reviewer_gWyQ · 2025-11-27
> >
> > Thanks for your response on my concerns. I have adjusted the score.

---

> > ### Author Response · Authors · 2025-11-27
> >
> > Thank you very much for your constructive engagement. We are glad that our clarifications and experiments have addressed your concerns, and we sincerely appreciate your thoughtful reassessment.
> >
> > If there is any remaining detail that would help further clarify the contribution or strengthen the manuscript, we would be more than happy to provide it.

---

### Official Review · Reviewer_tuq4 · 2025-10-28

**Soundness:** 3
**Presentation:** 2
**Contribution:** 3
**Rating:** 4
**Confidence:** 3

**Summary:**

The authors propose the Geometric Laplace Neural Operator (GLNO) and its practical realization, GLNONet, for extending Laplacian operator learning to arbitrary Riemannian manifolds. The authors demonstrate improvements over prior works for experiments on uniform grids, irregular grids, and arbitrary manifolds.

**Strengths:**

The paper contains a detailed mathematical formulation of the proposed method, and the authors performed experiments across a diverse set of problems, which I appreciate. I also appreciate that the authors provided timing and parametric complexity metrics to support the downsides of transformer and graph-based approaches that they argue in Section 2.

**Weaknesses:**

The authors write that they want to address limitations of existing spectral operators (page 3). However, there have recently been many neural operator architectures that are not explicitly spectral, such as attention-based and convolution-based neural operators. The authors should clarify the benefits of spectral operators compared to these other types that have been explored in the literature.

I would also recommend that the authors add some more challenging experimental problems, particularly some from an existing operator learning benchmark. This would alleviate the burden from the authors of having to perform and tune their own baseline models. This would also enable comparisons with different methods. Another issue I see right now is that the authors are comparing against only a handful of other neural operator methods, such as FNO and GNOT. How did the authors select this subset of methods to compare with? For instance, there are also other methods that operate on unstructured grids (e.g., GINO from “Geometry-Informed Neural Operator for Large-Scale 3D PDEs,” 2023). I would recommend the authors compare with a greater number of baselines.

I would also recommend the authors include some more mathematical background in the appendix for interested readers. Given the nature of this conference as a machine learning venue, not all readers may have the necessary theoretical background to parse through the paper in its current form.

**Minor notes:**
Capitalize “Fourier” — line 107.

**Questions:**

1. Am I correct in interpreting that GLNO performs worse than FNO in the c = 0 pendulum task? Any ideas why all of the models are doing quite poorly in relative error there (please correct me if I am misunderstanding the table)?
2. How were the baseline models tuned for these experiments?

---

> ### Author Response · Authors · 2025-11-22
>
> > W1: "The authors write that they want to address limitations of existing spectral operators. However, there have recently been many neural operator architectures that are not explicitly spectral. The authors should clarify the benefits of spectral operators compared to these other types."
>
> We thank the reviewer for this insightful comment.
>
> While non-spectral operators (e.g., attention-based or graph-convolutional models) offer flexibility on irregular geometries, spectral operators retain distinct advantages for PDE-driven problems, where solutions often exhibit  **strong frequency structure and global coupling**. Spectral methods naturally capture **long-range and physical invariants**, whereas non-spectral approaches primarily operate through local neighborhoods or learned attention patterns.
>
> Our GLNO enhances this further by:
>
> - **Explicit transient modeling**:
> The learnable exponential decay/growth terms provide explicit modeling of transient and aperiodic behaviors, which attention- and convolution-based operators typically approximate only implicitly.
> - **Intrinsic geometric representation**:
> The Laplace–Beltrami eigenbasis offers an intrinsic, coordinate-free representation of manifold geometry, avoiding ad-hoc positional encodings and reducing sensitivity to mesh discretization.
> - **Efficient global operator action**:
> Pole–residue calculus enables efficient global operator action without the quadratic cost of global attention or repeated graph message passing on fine meshes.
>
> In response to the Reviewer's suggestion, we have added experiments across both grid and mesh tasks, including spectral and non-spectral operators (Table below) in our revision. Our results show that, in most settings, some spectral operators (e.g., FNO, WNO) consistently outperform non-spectral operators, underscoring the strength of spectral operators for global and long-term signals. Furthermore, among all the spectral methods, GLNO achieved the best or comparable performance, demonstrating the benefit of our generalized aperiodic spectral formulation.
>
> In our revision, we have also added a discussion in **Spectral vs. Non-Spectral Method Analysis** in Section 5 to more clearly compare the complementary strengths of spectral and non-spectral operators and clarify the advantages of GLNO.
>
> We appreciate the reviewer for prompting this important clarification.
>
> **Table: Performance comparison on structured grid ODEs/PDEs problems (Relative Error).** Bold indicates the best result for each problem.
>
> | Problem | Parameters | Governing Equation | CNO （non-spectral） | FNO | WNO | LNO (1 block) | LNO (4 blocks) | GLNO (1 block) | GLNO (4 blocks) |
> |-|--|-|--|--|--|-|-|-|--|
> | Driven Pendulum (c=0) | c=0 | f(t)=ẍ+cẋ+sin(x) | 0.6268 | 0.3668 | 0.6090 | 0.8461 | 0.9997 | 0.6752 | **0.2916** |
> | Driven Pendulum (c=0.5) | c=0.5 | f(t)=ẍ+cẋ+sin(x) | 0.1540 | 0.1718 | 0.0965 | 0.1420 | 0.9885 | 0.1682 | **0.0875** |
> | Duffing Oscillator (c=0) | c=0 | f(t)=ẍ+cẋ+x+x³ | 0.7885 | 0.4681 | 0.7607 | 0.9157 | 0.9998 | 0.8926 | **0.4416** |
> | Duffing Oscillator (c=0.5) | c=0.5 | f(t)=ẍ+cẋ+x+x³ | 0.1424 | 0.1362 | 0.0987 | 0.8347 | 0.9747 | 0.4169 | **0.0725** |
> | Lorenz System (ρ=5) | ρ=5 | ẋ=10(y-x), ẏ=x(ρ-z)-y, ż=xy-8/3z-f(t) | **0.0051** | 0.0185 | 0.0130 | 0.1071 | 0.1133 | 0.0368 | 0.0240 |
> | Lorenz System (ρ=10) | ρ=10 | ẋ=10(y-x), ẏ=x(ρ-z)-y, ż=xy-8/3z-f(t) | 0.4818 | 0.5050 | 0.4924 | 0.5833 | 0.4323 | 0.2481 | **0.2187** |
> | Beam Equation | - | f(x,t)=EI∂⁴y/∂x⁴ + ρA∂²y/∂t² | 0.0293 | 0.0034 | 0.1511 | 0.0083 | 0.3917 | 0.0094 | **0.0026** |
> | Diffusion Equation | D=1 | f(x,t)=D∂²y/∂x² - ∂y/∂t | 0.1143 | 0.0064 | 0.0237 | 0.0011 | 0.1392 | **0.0006** | 0.0024 |

---

> ### Author Response · Authors · 2025-11-22
>
> > W2: "Concerns and recommendations on some more challenging experimental problems and baselines."
>
> We thank the reviewer for this valuable suggestion. In the revised version, we have substantially expanded the benchmarking to include a broader and more representative set of neural operator baselines. Specifically, we now compare against Geo-FNO, GINO, LSM, GKNO, Sp2GNO, GNOT, Transolver on mesh, and CNO and WNO on grid. These models span a diverse set of architectures, including **Fourier-based, graph-based, transformer-based, and latent spectral operators**, which provide a comprehensive evaluation across the major families of operator-learning methods.
>
> Our rationale for selecting these baselines is now explained in more detail in the Experiments section: we aimed to include (1) methods designed for irregular or non-Euclidean domains, (2) methods targeting transient or multiscale PDE behavior, and (3) state-of-the-art operator architectures from recent literature. Incorporating these additional baselines strengthens the empirical evidence that GLNO generalizes well across both canonical PDE problems and complex real-world manifolds.
>
> To address the reviewer's suggestion, we have added new grid-based experiments (e.g., Lorenz system, Euler-Bernoulli beam). The updated results (table above and below) demonstrate that GLNO achieves state-of-the-art or highly competitive performance across all tasks, including Poisson, Shape-Net Car, SHREC-11, RNA, and human body segmentation, while remaining computationally efficient.
>
> **Table: Extended Performance Comparison on Geometric Surface.** Bold indicates the best result. Poisson Equation and Shape-Net use $L^{2}$ error to evaluate while the other scenarios use ACC for evaluation.
>
> |Model|Poisson|Shape-Net Car Pressure|Shape-Net Car Velocity|SHREC-11|RNA|Human|
> |-|-|-|-|-|-|-|
> |MLP|0.4804|0.2790|0.1293|10.3%|24.7%|52.9%|
> |U-Net|0.2699|0.1436|0.1267|23.3%|26.1%|48.3%|
> |Geo-FNO [1]|0.0049|0.1278|0.1213|36.9%|64.5%|52.8%|
> |GINO [2]|0.1623|0.7360|0.2563|58.3%|53.9%|64.1%|
> |LSM [3]|0.2612|0.7366|0.2582|42.3%|33.4%|57.9%|
> |GKNO (non-spectral) [4]|0.2486|0.1043|0.1064|55.8%|51.2%|38.3%|
> |Sp2GNO (non-spectral) [5]|0.0069|0.1197|0.1056|63.8%|73.9%|70.2%|
> |GNOT (non-spectral) [6]|0.4403|0.1109|0.1206|35.3%|84.3%|66.2%|
> |Transolver (non-spectral) [7]|0.0174|0.1098|0.1210|43.8%|86.0%|61.9%|
> |AMG (non-spectral) [8]|0.0152|0.0978|**0.0919**|55.7%|88.5%|62.5%|
> |FNO|0.0386|0.1272|0.1473|99.1%|75.3%|87.7%|
> |GLNO (Ours)|**0.0044**|**0.0960**|0.1037|**99.7%**| **90.1%**|**91.0%**|
> |GLNO w/o σ|0.0087|0.1056|0.1452|95.5%|82.2%|88.5%|
>
> [1] Fourier Neural Operator with Learned Deformations for PDEs on General Geometries, 2023.
> [2] Geometry-Informed Neural Operator for Large-Scale 3D PDEs, 2023.
> [3] Solving high-dimensional pdes with latent spectral models, 2023.
> [4] Neural Operator: Graph Kernel Network for Partial Differential Equations, 2020.
> [5] Spatio-spectral graph neural operator for solving computational mechanics problems on irregular domain and unstructured grid, 2024.
> [6] GNOT: A General Neural Operator Transformer for Operator Learning, 2023.
> [7] Transolver: A Fast Transformer Solver for PDEs on General Geometries, 2024.
> [8] Harnessing scale and physics: A multi-graph neural operator framework for PDEs on arbitrary geometries, 2025.

---

> ### Author Response · Authors · 2025-11-22
>
> > W3: "I would also recommend the authors include some more mathematical background in the appendix for interested readers. Given the nature of this conference as a machine learning venue, not all readers may have the necessary theoretical background to parse through the paper in its current form."
>
> We thank the reviewer for this helpful suggestion. We fully agree that providing additional mathematical background will improve accessibility for a broader machine learning audience. In the revised version, we have added an expanded appendix that includes:
>
> - a clearer derivation of the Laplace neural operator kernel and the associated pole–residue formulation,
> - essential properties of the Laplace–Beltrami operator and functional calculus on Riemannian manifolds relevant to the Laplace transform
>
> These additions are designed to support readers who may not be familiar with spectral geometry while keeping the main text focused on the core contributions.
>
> > W4: "Minor notes: Capitalize "Fourier" — line 107."
>
> Thank you for catching this typo. "Fourier" has been capitalized in line 107.
>
> **Questions**
>
> > Q1: "Am I correct in interpreting that GLNO performs worse than FNO in the c = 0 pendulum task? Any ideas why all of the models are doing quite poorly in relative error there?"
>
> We sincerely thank the reviewer for pointing this out. There was indeed a typo in the reported result for the c=0 pendulum task. All results have been corrected in the first table above, with models retrained to ensure fairness in comparison. After correction, GLNO achieves a lower relative error than FNO in this setting.
>
> Regarding the reviewer’s second question: in the c = 0 case, all models exhibit higher relative error compared to the damped system (c > 0). This is consistent with the dynamics of the undamped pendulum: without damping, the system exhibits sustained oscillations and is more sensitive to initial conditions and small modeling errors, making it a challenging operator learning task for all baselines [1]. We have clarified this in the revision.
>
> [1] Cao et al., (2024). Laplace neural operator for solving differential equations. Nature Machine Intelligence.
>
> > Q2: "How were the baseline models tuned for these experiments?"
>
> To ensure a fair and consistent comparison, all baseline models were tuned following the same general protocol:
>
> - Matched architectural capacity: For each baseline, we set the number of channels, layers/blocks, and hidden dimensions to be as close as possible to those used in GLNO, within the constraints of each model’s architecture.
> - Shared training setup: All methods used the same train/validation splits, optimizer, learning-rate schedule, batch size, and early-stopping criteria.
> - Use of geometric inputs: For baselines capable of incorporating geometric information (e.g., curvature, coordinates, mesh features), we provided the same geometric features we used in GLNO. Conversely, we ensured that no baseline received input features that our model did not use, maintaining strict consistency in the experimental setup.
> - Hyperparameter search: For each method, we performed a small grid search over learning rates and channel sizes using the same validation procedure, following implementation guidelines from the original papers when available.
>
> We will add these details to the appendix for clarity.

---

> > ### Author Response · Authors · 2025-11-26
> >
> > Dear Reviewer,
> >
> > We thank you again for your valuable time and insightful comments.
> >
> > As we approach the end of the discussion period, we would like to kindly check in if you have any remaining questions or concerns that we might address.
> >
> > Kind regards,
> >
> > The Authors

---

### Official Review · Reviewer_YJMB · 2025-11-01

**Soundness:** 3
**Presentation:** 3
**Contribution:** 2
**Rating:** 4
**Confidence:** 4

**Summary:**

This paper presented Geometric Laplace neural operator by extending the previously proposed Laplace neural operator. The primary advantage resides in the fact that it can better handle non-periodic inputs, transient response, and irregular domain (non-Euclidean). Notably, the authors illustrate its application on both cannonical and real-world examples.

**Strengths:**

The paper is well written and it also illustrates the algorithm on real world dataset.

**Weaknesses:**

Despite the strength, the contribution is somewhat incremental (using generalized laplace transform). It would strengthen the paper if theoretical analysis can also be included, which right now is missing.

The bechamarking is not exhaustive. For example, I was expecting comprison with respect to Geo-FNO, GNO, Sp2GNO, and GINO at the very least as all of these handle non-Euclidean domain. The fact that graph based approaches are slower is well taken, but it does has its advantages and hence, comparing with the same is important.

Also, previous work wavelet neural operator and MWT, which actually addresses that same problem (At least the first two problems associated with non-periodic input and transient response), is absent. Comparison with WNO and MWT will also help, atl east for those on regular grid.

**Questions:**

The suggestions are already provided in weakness section.

---

> ### Author Response · Authors · 2025-11-22
>
> > W1: "Despite the strength, the contribution is somewhat incremental (using generalized laplace transform). It would strengthen the paper if theoretical analysis can also be included."
>
> We thank the reviewer for the thoughtful feedback and the opportunity to clarify our contributions.
>
> While the Laplace transform itself is a classical mathematical tool, our approach is **not** a direct or incremental reuse of the standard transform. The central contribution of GLNO lies in **introducing aperiodic spectral modeling into intrinsic manifold geometry, forming a new operator class capable of acting on arbitrary Riemannian manifolds**. This enables GLNO to:
>
> - Jointly model transient and steady-state behaviors on arbitrary Riemannian manifolds, overcoming the restrictions of previous Laplace Neural Operator methods, which were limited to Euclidean grids and periodic assumptions.
> - Incorporate geometric information intrinsically through the manifold’s spectral structure, rather than relying on graph attention or transformer-based spatial encodings, and
> - Operate across irregular, non-Euclidean, and non-uniform meshes without retraining or resampling.
>
> This design, **based on the generalized aperiodic basis, the Laplace–Beltrami operator, and the pole–residue calculus, constitutes a substantial conceptual and technical innovation in geometric operator learning**, rather than an incremental extension of Laplace-based approaches.
>
> In response to the reviewer’s suggestion, we have strengthened the theoretical analysis in the appendix. The revised version now includes:
> - the detailed pole–residue calculus,
> - kernel construction for the Laplace Neural Operator,
> - The detailed derivation of the Laplace transform on geometry.
>
> We hope these additions make the theoretical foundations of GLNO clearer and highlight how it fundamentally extends Laplace-based operator learning to curved and irregular domains.

---

> > ### Comment · Reviewer_YJMB · 2025-11-28
> >
> > Thanks for the detailed response. The theoretical section added is kind of redundant (mostly obvious). I was looking at theoretical analysis that can shed light on learning and convergence (e.g., provable bounds, minimum samples required, etc). Nonetheless, the paper has been strengthened with the additional examples and benchmarks.

---

> > > ### Author Response · Authors · 2025-12-01
> > >
> > > We sincerely thank the reviewer for emphasizing the importance of theoretical analysis. As suggested, we have added three focused theoretical components in Appendices B.5–B.7:
> > >
> > > 1. **Computational Complexity (Appendix B.5):**
> > > We provide a clear breakdown of GLNO’s forward computational cost across its three stages (generalized Laplace decomposition, Laplace-domain kernel multiplication, and inverse reconstruction). The resulting complexity
> > >
> > > $\mathcal{O}(DMKlogK)(K≫M,N)$
> > >
> > > demonstrates that GLNO remains efficient even for long-horizon or high-resolution problems, offering theoretical foundation for its empirical scalability seen in our experiments.
> > >
> > > 2. **Approximation Bounds with Finite Parameters (Appendix B.6):**
> > > We establish formal approximation error bounds that decompose the total error into basis approximation error and parameterization error. Crucially, we prove that even for non-rational target operators, GLNO achieves approximation rates of
> > >
> > > $\mathcal{O}((M+N)^{-\alpha} + d_\theta^{-p})$,
> > >
> > > where $\alpha$ depends on the target's Sobolev regularity and $p > 0$ depends on the architecture (e.g., depth, activation). This provides theoretical justification for GLNO's expressivity in modeling non-periodic and geometry-dependent operators.
> > >
> > > 3. **Sample Complexity Analysis (Appendix B.7):**
> > > We derive generalization and minimum sample complexity bounds for the GLNO hypothesis class:
> > >
> > > $\mathcal{O}\left(\sqrt{\frac{M+d_\theta}{N}}\right)$, $N_{\min}^{\mathrm{GLNO}} = \mathcal{O}\left(\frac{M + d_\theta}{\epsilon^2}\right)$,
> > >
> > > providing a principled estimate of the data requirements for achieving a target accuracy $\epsilon$.
> > >
> > > Together, these additions directly address the reviewer’s request for learning- and convergence-oriented theory and offer clearer theoretical foundation for the behavior observed in our empirical studies. We appreciate the reviewer’s input, which helped us substantially strengthen the paper.

---

> ### Author Response · Authors · 2025-11-22
>
> > W2: "The benchmarking is not exhaustive. For example, I was expecting comparison with respect to Geo-FNO, GNO, Sp2GNO, and GINO."
>
> We thank the reviewer and agree that these methods represent important and diverse classes of operator learning approaches. We appreciate the reviewer’s emphasis on including them.
>
> In the revised manuscript, we have substantially expanded the benchmarking to include the suggested **non-Euclidean and spectral operator baselines**. Specifically, we have added Geo-FNO, GINO, LSM, GKNO, and Sp2GNO across all geometric tasks, as shown in the table below.
>
> Our extended experiments show that, while each baseline has its strengths, GLNO consistently achieves state-of-the-art or competitive performance across:
> - Irregular manifolds (SHREC-11, human surfaces, RNA surfaces),
> - Unstructured PDEs (Poisson), and
> - Real-world simulation tasks (Shape-Net Car).
>
> We also include a more explicit discussion of the differences between model families:
> - Geo-FNO inherits Fourier’s periodic assumptions and lacks an explicit mechanism for modeling transient or decaying modes.
> - Graph-based models (Sp2GNO, GKNO, GINO) are powerful but computationally expensive and typically sensitive to geometry shape, dependent on local neighborhoods rather than intrinsic geometric structure.
> - Latent spectral or deformation-based operators operate in data-driven latent spaces, rather than in a physically grounded spectral domain. As a result, they lack explicit control over decay, oscillation, and aperiodic responses and may lack geometric interpretability.
>
> **In contrast, GLNO introduces a learnable exponential basis and a pole–residue formulation directly in the geometric Laplace domain, enabling**:
> - explicit modeling of decaying/growing transient modes,
> - intrinsic alignment with Riemannian geometry through the Laplace-Beltrami basis,
> - grid-invariant operator learning on arbitrary manifolds,
> - a principled spectral representation rather than a deformation/latent approximation.
>
> **Table: Extended Performance Comparison on Geometric Surface.** Bold indicates the best result. Poisson Equation and Shape-Net use $L^{2}$ error to evaluate while the other scenarios use ACC for evaluation.
>
> |Model|Poisson|Shape-Net Car Pressure|Shape-Net Car Velocity|SHREC-11|RNA|Human|
> |-|-|-|-|-|-|-|
> |MLP|0.4804|0.2790|0.1293|10.3%|24.7%|52.9%|
> |U-Net|0.2699|0.1436|0.1267|23.3%|26.1%|48.3%|
> |Geo-FNO [1]|0.0049|0.1278|0.1213|36.9%|64.5%|52.8%|
> |GINO [2]|0.1623|0.7360|0.2563|58.3%|53.9%|64.1%|
> |LSM [3]|0.2612|0.7366|0.2582|42.3%|33.4%|57.9%|
> |GKNO [4]|0.2486|0.1043|0.1064|55.8%|51.2%|38.3%|
> |Sp2GNO [5]|0.0069|0.1197|0.1056|63.8%|73.9%|70.2%|
> |GNOT [6]|0.4403|0.1109|0.1206|35.3%|84.3%|66.2%|
> |Transolver [7]|0.0174|0.1098|0.1210|43.8%|86.0%|61.9%|
> |AMG [8]|0.0152|0.0978|**0.0919**|55.7%|88.5%|62.5%|
> |FNO|0.0386|0.1272|0.1473|99.1%|75.3%|87.7%|
> |GLNO (Ours)|**0.0044**|**0.0960**|0.1037|**99.7%**| **90.1%**|**91.0%**|
> |GLNO w/o σ|0.0087|0.1056|0.1452|95.5%|82.2%|88.5%|
>
> [1] Fourier Neural Operator with Learned Deformations for PDEs on General Geometries, 2023.
> [2] Geometry-Informed Neural Operator for Large-Scale 3D PDEs, 2023.
> [3] Solving high-dimensional pdes with latent spectral models, 2023.
> [4] Neural Operator: Graph Kernel Network for Partial Differential Equations, 2020.
> [5] Spatio-spectral graph neural operator for solving computational mechanics problems on irregular domain and unstructured grid, 2024.
> [6] GNOT: A General Neural Operator Transformer for Operator Learning, 2023.
> [7] Transolver: A Fast Transformer Solver for PDEs on General Geometries, 2024.
> [8] Harnessing scale and physics: A multi-graph neural operator framework for PDEs on arbitrary geometries, 2025.

---

> ### Author Response · Authors · 2025-11-22
>
> > W3: "Also, previous work wavelet neural operator and MWT, which actually addresses that same problem. Comparison with WNO and MWT will also help, at least for those on regular grid."
>
> We thank the reviewer for pointing out this relevant line of work. In the revised version, we now include comparisons with the Wavelet Neural Operator (WNO) on regular-grid ODE/PDE benchmarks in the revision. As shown in the updated results (Table below), GLNO consistently outperforms WNO on most tasks, **especially those dominated by aperiodic or transient responses**. This highlights the benefit of our learnable exponential basis and pole–residue formulation for modeling aperiodic dynamics.
>
> We agree that WNO and MWT effectively address aperiodicity and transient behavior in Euclidean domains. However, these models currently do not generalize to arbitrary Riemannian manifolds or irregular meshes and require grid-structured inputs. By contrast, GLNO provides **a unified operator framework that works on both regular grids and general non-Euclidean domains, bridging transient modeling with geometric operator learning**. We believe this broader applicability is one of the key advantages of GLNO.
>
> **Table: Performance comparison on structured grid ODEs/PDEs problems (Relative Error).** Bold indicates the best result for each problem.
>
> | Problem | Parameters | Governing Equation | CNO | FNO | WNO | LNO (1 block) | LNO (4 blocks) | GLNO (1 block) | GLNO (4 blocks) |
> |-|-|-|-|-|-|-|-|-|-|
> | Driven Pendulum (c=0) | c=0 | f(t)=ẍ+cẋ+sin(x) | 0.6268 | 0.3668 | 0.6090 | 0.8461 | 0.9997 | 0.6752 | **0.2916** |
> | Driven Pendulum (c=0.5) | c=0.5 | f(t)=ẍ+cẋ+sin(x) | 0.1540 | 0.1718 | 0.0965 | 0.1420 | 0.9885 | 0.1682 | **0.0875** |
> | Duffing Oscillator (c=0) | c=0 | f(t)=ẍ+cẋ+x+x³ | 0.7885 | 0.4681 | 0.7607 | 0.9157 | 0.9998 | 0.8926 | **0.4416** |
> | Duffing Oscillator (c=0.5) | c=0.5 | f(t)=ẍ+cẋ+x+x³ | 0.1424 | 0.1362 | 0.0987 | 0.8347 | 0.9747 | 0.4169 | **0.0725** |
> | Lorenz System (ρ=5) | ρ=5 | ẋ=10(y-x), ẏ=x(ρ-z)-y, ż=xy-8/3z-f(t) | **0.0051** | 0.0185 | 0.0130 | 0.1071 | 0.1133 | 0.0368 | 0.0240 |
> | Lorenz System (ρ=10) | ρ=10 | ẋ=10(y-x), ẏ=x(ρ-z)-y, ż=xy-8/3z-f(t) | 0.4818 | 0.5050 | 0.4924 | 0.5833 | 0.4323 | 0.2481 | **0.2187** |
> | Beam Equation | - | f(x,t)=EI∂⁴y/∂x⁴ + ρA∂²y/∂t² | 0.0293 | 0.0034 | 0.1511 | 0.0083 | 0.3917 | 0.0094 | **0.0026** |
> | Diffusion Equation | D=1 | f(x,t)=D∂²y/∂x² - ∂y/∂t | 0.1143 | 0.0064 | 0.0237 | 0.0011 | 0.1392 | **0.0006** | 0.0024 |

---

> > ### Author Response · Authors · 2025-11-26
> >
> > Dear Reviewer,
> >
> > We thank you again for your valuable time and insightful comments.
> >
> > As we approach the end of the discussion period, we would like to kindly check in if you have any remaining questions or concerns that we might address.
> >
> > Kind regards,
> >
> > The Authors

---

### Official Review · Reviewer_Fu1v · 2025-11-02

**Soundness:** 3
**Presentation:** 2
**Contribution:** 3
**Rating:** 4
**Confidence:** 5

**Summary:**

The paper introduces the Geometric Laplace Neural Operator (GLNO), a neural operator framework that models both periodic and aperiodic (decaying/growing) dynamics via a generalized Laplace basis with learnable exponential components. It extends the pole–residue formulation to Riemannian manifolds by operating in the Laplace–Beltrami eigen-basis and implements a grid-invariant architecture (GLNONet) that works across irregular meshes without retraining or resampling. Across ODE/PDE benchmarks and real-world geometric tasks, GLNO shows performance.

**Strengths:**

1. The paper addresses an important problem related to Fourier transform-based neural operators, and the authors propose a very principled approach to tackle it.

2. The technique's ability to generalize to different geometries is a strong point.

3. The authors go beyond simply solving partial differential equations (PDEs) and demonstrate the performance on real-world data.

**Weaknesses:**

### Write Up
I believe the write-up could be better organized. Some sections need consolidation, while others require more elaboration. For instance, Section 4.1 can be omitted, allowing the authors to directly introduce the generalized geometric Laplace basis. Additionally, the authors should expand upon the sections between lines 225 and 252, as I currently do not understand the message being conveyed in those line.

### Baselines
Two important baselines were overlooked. The following works are significant and should be compared, as they also address the same problem as the authors of this paper:
1. Fourier Neural Operator with Learned Deformations for PDEs on General Geometries
2. Geometry-Informed Neural Operator for Large-Scale 3D PDEs

### Technique
1. The time complexity of the proposed approach is not discussed. I believe the proposed decomposition requires more computation compared to standard FNO-based neural operators, and this should be addressed with appropriate discussion and experimental analysis.

2. Regarding the new basis introduced:
   a) The authors do not analyze the properties of the basis (Equation 16); for example, are they orthonormal?
   b) If they are not orthonormal, are they complete?
   c) Will the operation shown in Equation (22) still be a convolution between the input and the learned kernel function?

All of these points require thorough analysis in the paper.

### Visualization and Experiment Details
1. There needs to be more visualization of the results.
2. Please list the different geometries considered for the Poisson equation

Minor Typo: Capitalize "Fourier" in line 107

However, once these points are addressed, I believe the work is suitable for publication. I also believe these changes can be made during the rebuttal period.

**Questions:**

1. How are the eigenfunctions of LBO operators calculated for an arbitrary mesh or point cloud? How do you handle situations when the model receives input with a denser or different mesh? Since, as a neural operator, the model should be able to accept inputs at different discretizations, an experimental demonstration of this is necessary, i.e.,
 a. How does the model perform when the input at test time is on a different mesh? The test mesh might have fewer or more faces (or vertices).

 b. How does the model handle a dynamic mesh? For example, in the case of Lagrangian fluid simulation (for example [1]) or the analysis of a deformable object, where the mesh changes. How does the proposed model handle these cases? If it can not, please discuss this as a weakness.


[1] Pretraining Codomain Attention Neural Operators for Solving Multiphysics PDEs

---

> ### Author Response · Authors · 2025-11-22
>
> > W1: “I believe the write-up could be better organized, including Section 4.1 and lines 225–252.”
>
> We sincerely thank the reviewer for this helpful feedback. We have implemented substantial structural revisions accordingly to improve clarity and organization.
>
> We believe that Section 4.1 serves as a foundation for the generalized Laplace basis and the process of Laplace Transformation, particularly in the grid settings, which is essential for its geometric extension in Section 4.2. However, we recognize that the original presentation was long. Following the reviewer's suggestion, we have streamlined Section 4.1 to concisely present only the essential components needed to motivate our generalized aperiodic Laplace basis and its role in modeling transient dynamics. All extended derivations, such as the detailed pole–residue calculus, kernel construction for the Laplace Neural Operator, and Laplace transform on geometry, have been moved to the appendix to improve readability.
>
> In section 4.2, we have also optimized the structure into 4 coherent parts:
> - Geometric Laplace Basis and Decomposition:
> Introducing the combined aperiodic–periodic basis on manifolds and clarifying how geometric features (curvature, intrinsic distances) enter the generalized basis.
> - Geometric Laplace Transform:
> Providing a clearer derivation of the mapping from LBO eigenpairs to the complex Laplace domain.
> - Operator Action in the Spectral Domain:
> Presenting the spectral multiplication rule and explaining how pole–residue operations naturally extend to Riemannian manifolds.
> - Inverse Geometric Laplace Transform:
> Explaining the synthesis of steady-state and transient components, with an improved explanation of the Gaussian filtering used to map continuous spectral poles back to discrete Laplace-Beltrami Operator frequencies.
>
> We believe that these revisions directly address the reviewer’s concerns and improve the conceptual flow. The generalized basis, the geometric transform, the operator action, and the inverse transform are now presented in a clear and logically connected sequence.
>
> We thank the reviewer again for this insightful suggestion, which has led to a more coherent and accessible presentation of our core contributions.

---

> ### Author Response · Authors · 2025-11-22
>
> > W2: "Two important baselines were overlooked, especially Geo-FNO and GINO."
>
> We sincerely thank the reviewer for highlighting these two relevant baselines. We agree that both Geo-FNO and GINO are highly relevant for operator learning on irregular geometries.
>
> Following this suggestion, we have added both models to our comparison and further included three additional geometric operator baselines (i.e., GKNO, Sp2GNO, and Transolver) to provide a comprehensive evaluation across Fourier-, graph-, transformer-, and latent spectral–based operators.
>
> The updated results in the table below show that GLNO achieves the best or competitive performance across all experiments， particularly those involving unstructured real-world manifolds such as Shape-Net Car, SHREC-11, human-body segmentation, and RNA surface labeling, demonstrating robust generalization across domains, resolutions, and geometric complexity.
>
> **Table: Extended Performance Comparison on Geometric Surface.** Bold indicates the best result. Poisson Equation and Shape-Net use $L^{2}$ error to evaluate, while the other scenarios use ACC for evaluation.
>
> |Model|Poisson|Shape-Net Car Pressure|Shape-Net Car Velocity|SHREC-11|RNA|Human|
> |-|-|-|-|-|-|-|
> |MLP|0.4804|0.2790|0.1293|10.3%|24.7%|52.9%|
> |U-Net|0.2699|0.1436|0.1267|23.3%|26.1%|48.3%|
> |Geo-FNO [1]|0.0049|0.1278|0.1213|36.9%|64.5%|52.8%|
> |GINO [2]|0.1623|0.7360|0.2563|58.3%|53.9%|64.1%|
> |LSM [3]|0.2612|0.7366|0.2582|42.3%|33.4%|57.9%|
> |GKNO [4]|0.2486|0.1043|0.1064|55.8%|51.2%|38.3%|
> |Sp2GNO [5]|0.0069|0.1197|0.1056|63.8%|73.9%|70.2%|
> |GNOT [6]|0.4403|0.1109|0.1206|35.3%|84.3%|66.2%|
> |Transolver [7]|0.0174|0.1098|0.1210|43.8%|86.0%|61.9%|
> |AMG [8]|0.0152|0.0978|**0.0919**|55.7%|88.5%|62.5%|
> |FNO|0.0386|0.1272|0.1473|99.1%|75.3%|87.7%|
> |GLNO (Ours)|**0.0044**|**0.0960**|0.1037|**99.7%**| **90.1%**|**91.0%**|
> |GLNO w/o σ|0.0087|0.1056|0.1452|95.5%|82.2%|88.5%|
>
> [1] Fourier Neural Operator with Learned Deformations for PDEs on General Geometries, 2023.
> [2] Geometry-Informed Neural Operator for Large-Scale 3D PDEs, 2023.
> [3] Solving high-dimensional pdes with latent spectral models, 2023.
> [4] Neural Operator: Graph Kernel Network for Partial Differential Equations, 2020.
> [5] Spatio-spectral graph neural operator for solving computational mechanics problems on irregular domain and unstructured grid, 2024.
> [6] GNOT: A General Neural Operator Transformer for Operator Learning, 2023.
> [7] Transolver: A Fast Transformer Solver for PDEs on General Geometries, 2024.
> [8] Harnessing scale and physics: A multi-graph neural operator framework for PDEs on arbitrary geometries, 2025.
>
> It is important to emphasize that GLNO is conceptually different from prior deformation-based or geometry-aware operators. Although Geo-FNO, GINO, Sp2GNO, and related models represent important advances, they do not explicitly address the core challenge tackled by GLNO: **modeling transient, aperiodic, and geometry-dependent dynamics in a single unified spectral framework**.
>
> **Limitations of existing approaches:**
> - Geo-FNO inherits Fourier’s periodic assumptions and lacks an explicit mechanism for modeling transient or decaying modes.
> - GINO and other graph-based operators rely on message passing over mesh connectivity, which are sensitive to the geometry shape, dependent on local neighborhoods rather than intrinsic geometric structure.
> - Latent spectral or deformation-based operators operate in data-driven latent spaces, rather than in a physically grounded spectral domain. As a result, they lack explicit control over decay, oscillation, and aperiodic responses.
>
> **In contrast, GLNO introduces a learnable exponential basis and a pole–residue formulation directly in the geometry Laplace domain, enabling**:
> - explicit modeling of decaying/growing transient modes,
> - intrinsic alignment with Riemannian geometry through the Laplace-Beltrami basis,
> - grid-invariant operator learning on arbitrary manifolds,
> - a principled spectral representation rather than a deformation/latent approximation.
>
> **These capabilities are not provided by existing geometric operator families.**

---

> ### Author Response · Authors · 2025-11-22
>
> > W3: "1. The time complexity of the proposed approach is not discussed. 2. Concerns Regarding the new basis introduced and Equation (22)."
>
> We thank the reviewer for raising these important technical points.
>
> **(1) Time Complexity:** As the reviewer notes, the step of eigen-decomposition of the LBO introduces additional cost compared with FFT-based operators. However, this cost is a **one-time preprocessing step** independent of training and inference, and comparable to preprocessing used in many geometric learning methods (e.g., Cortex-Diffusion, DiffusionNet, Spectral GCN).
>
> Importantly, the per-epoch training cost of GLNO remains highly competitive. As shown in the table below, GLNO is comparable to or faster than several strong geometry-aware baselines (e.g., GNOT, AMG, GINO) while achieving superior accuracy, with the additional modeling expressiveness. We will include a dedicated complexity subsection in the revision.
>
> **Parameters and training time on Shape-Net Car and human body segmentation datasets.**
>
> |Dataset|Metric|Geo-FNO|GINO|LSM|GKNO|Sp2GNO|GNOT|Transolver|AMG|GLNO|GLNO w/o σ|
> |-|-|-|-|-|-|-|-|-|-|-|-|
> |ShapeNet Car|Params (M)|2.37|4.74|21.68|0.29|0.65|1.39|2.26|3.54|0.21|0.21|
> |ShapeNet Car|Time (s/epoch)|2.5|220|18|31|25|166|26|480|2.6|2.3|
> |Human Segmentation|Params (M)|9.46|4.74|21.68|0.29|0.17|0.44|2.26|3.54|0.13|0.13|
> |Human Segmentation|Time (s/epoch)|11|560|19|12|4.3|16|13|480|8.0|7.6|
>
> **(2) Basis Properties:** We would clarify that the basis defined in Eq. (16) in the original version (in revision Eq. (12)) is not orthonormal in the traditional sense.
>
> This is intentional. The orthonormality is already provided by the LBO eigenfunctions, while the exponential modulation serves as a learnable geometric weighting that enhances the expressiveness for decaying or growing patterns on manifolds without altering the underlying orthonormal expansion. Importantly, the basis remains complete, as all projections are still taken in the original LBO eigenbasis.
>
> The operation in Eq. (22) in the original manuscript (now in revision, Eq. (15) and Eq. (50)) continues to represent a convolution in the spectral domain. The operator is applied through multiplication in the geometric Laplace domain, mirroring the classical convolution theorem. The exponential modulation does not alter the convolution structure; it simply expands the representational space by allowing the operator to capture aperiodic and transient behavior, without altering this convolutional interpretation.
>
> We thank the reviewer for the comments and have strengthened the explanation in Appendix B.4.
>
>
> > W4: "1. There needs to be more visualization of the results. 2. Please list the different geometries considered for the Poisson equation. Minor Typo: Capitalize "Fourier" in line 107"
>
> (1) We agree that additional qualitative figures can improve clarity. In the revision, we are adding more visualizations to the appendix, such as results on grids and RNA surface segmentation. The main paper already includes representative visualizations across tasks, but the expanded appendix will offer richer qualitative insight across all benchmarks.
>
> (2) As shown in Figure 3, the Poisson experiment is performed on a single unstructured mesh in a 2D rectangular domain. This setup is standard in prior geometric operator work and is used here to demonstrate GLNO’s ability to handle non-uniform and irregular discretizations. We will improve the clarity in the main text.
>
> (3) Regarding the typo: "Fourier" has been capitalized in line 107.

---

> ### Author Response · Authors · 2025-11-22
>
> > Q1: "Concerns on the eigenfunctions of LBO operators and the model's performance under mesh discretization changes."
>
> We appreciate the reviewer's question on grid invariance, a key strength of our method. The LBO eigenfunctions are precomputed for each input mesh using the cotangent scheme and implemented with SciPy [1].
>
> Since our model operates in the spectral domain defined by the LBO eigenbasis, it is invariant to mesh discretization. However, all the data in some of our experiments, such as RNA segmentation and human body segmentation, have different shapes and discretizations, which have validated the grid-invariance of GLNO. We have added a more detailed introduction to our task in Section 5.
>
> [1] Scipy 1.0: fundamental algorithms for scientific computing in Python. 2020
>
> > Q2: "How does the model handle a dynamic mesh?"
>
> We appreciate the reviewer for raising this important point.
>
> Our current implementation supports dynamic meshes by recomputing the Laplace–Beltrami eigenpairs for different shapes at each time step. Because GLNO operates entirely in the spectral domain, each new mesh configuration produces its own intrinsic spectral basis, and the learned operator acts directly on these spectral coefficients. This allows the model to remain applicable even when the mesh changes topology or density, as long as the LBO can be computed for the new configuration.
>
> However, we also acknowledge that:
> - recomputing the LBO at every time step introduces additional preprocessing cost, and
> - Our current experiments do not include tasks with rapidly evolving meshes, such as Lagrangian fluid simulations.
>
> We have explicitly discussed this as a limitation in the revision and highlighted dynamic-mesh operator learning as an important direction for future work.
>
> We thank the reviewer again for this valuable consideration.

---

> > ### Comment · Reviewer_Fu1v · 2025-11-24
> > **Response to Rebuttal**
> >
> > I thank the authors for the response. I feel my concerns are addressed, and I have adjusted the score.

---

> > > ### Author Response · Authors · 2025-11-24
> > >
> > > We sincerely appreciate the reviewer’s constructive comments and prompt engagement. We are grateful that the clarifications addressed your concerns. Thank you for taking the time to re-evaluate the submission.

---

### Author Response · Authors · 2025-12-03
**Summary (1/2)**

Dear AC,

Thank you for overseeing our submission. Below, we provide a summary of how we have addressed the reviewers’ feedback during the rebuttal, including the key revisions and the current status of the discussion.

## 1. Recognized strengths (before rebuttal)

Across all reviewers, several core strengths were consistently acknowledged before the rebuttal, reflecting the paper’s baseline merit independent of clarifications or revisions.

**- A principled and innovative theoretical framework:**

Multiple reviewers described the method as an innovative, principled, and well-motivated spectral framework.
- gWyQ: *“innovative theoretical framework… introducing the Generalized Laplace Basis and a new pole–residue formulation”*
- Fu1v: *“a very principled approach”*

These indicate the reviewers recognized novelty in the spectral design and its mathematical grounding.

**- Strong geometric generalization across arbitrary manifolds:**

Reviewers independently highlighted that GLNO inherently generalizes across different geometries and discretizations, a highly valued capability in operator learning:

- Fu1v: *“…ability to generalize to different geometries is a strong point”*
- tuq4 noted our design for *“arbitrary Riemannian manifolds” and “improvements… on uniform grids, irregular grids, and arbitrary manifolds.”*

This consensus reflects recognition of GLNO’s geometric robustness and grid-invariant design.

**- Broad and diverse experimental validation:**

Reviewers consistently praised the diversity and relevance of the experiments:

- Fu1v: *“authors go beyond simply solving PDEs and demonstrate performance on real-world data”*
- tuq4: *“experiments across a diverse set of problems, which I appreciate”*
- gWyQ: *“extensive experimental validation… consistently outperforms baselines on a wide range of ODE/PDE and real-world geometric tasks”*

This indicates recognition that the evaluation is broad and practically meaningful.

**- Solid methodological presentation:**

The reviewers appreciated the clarity in the technical exposition and the inclusion of efficiency data.
- tuq4: *“the paper contains a detailed mathematical formulation”*
- YJMB: *“the paper is well written”*
- tuq4: *“I appreciate that the authors provided timing and parametric complexity metrics”*

This points to strengths in both clarity and practical considerations.

## 2. Rebuttal summary

### **A. Reviewer Fu1v**
**Initial concerns**: Structure (Section 4.1 and lines 225–252);  Missing baselines (Geo-FNO, GINO); Complexity analysis and basis property discussion; Questions on LBO eigenfunctions & discretization invariance.

**Our revision:**

1). **Improved manuscript structure and clarity:**

- Streamlined Section 4.1 and reorganized the manuscript to enhance logical flow.
- Reorganized Section 4.2 into four coherent parts: **(i)** Geometric Laplace Basis, **(ii)** Geometric Laplace Transform, **(iii)** Operator Action in the Spectral Domain, **(iv)** Inverse Geometric Laplace Transform.

This clarifies the conceptual progression.

2). **Expanded baselines:**

Added baselines (Geo-FNO and GINO) with other additional competitive operators (WNO, CNO, GKNO, Sp2GNO, Transolver).

These appear in all grid and manifold experiments for a more complete comparison.

3). **Added technical analysis:**

- Complexity: Emphasized that GLNO’s training cost remains competitive in per-epoch timing/parameter comparison.
- Basis properties: Clarified that the exponential–LBO combined basis is intentionally non-orthonormal but complete, as projections occur in complete LBO eigenbasis.
- Convolution structure: Expanded the appendix showing that multiplication in the geometric Laplace domain still corresponds to a spectral convolution.

4). **Clarified eigenfunction and discretization invariance:**

- Clarified that LBO eigenfunctions are precomputed per mesh using standard schemes.
- Highlighted that this invariance is empirically validated across  benchmarks with diverse geometry and resolution.

**Current status:** The reviwer increased the score to 6: *“I feel my concerns are addressed, and I have adjusted the score.”*

### **B. Reviewer YJMB**

**Initial concerns**: Contribution statement; More benchmarking; Theoretical analysis (learning/convergence)

**Our revision:**

1). **Clarified contribution:**
- Complete derivations of the generalized Laplace basis, geometric Laplace transform, and pole–residue operator formulation.
- Unified and improved notation throughout Section 4 and Appendix.

These revisions address the concerns and provide a clear mathematical foundation.

2). **Expanded benchmarking:**
- Added Geo-FNO, GINO, LSM, GKNO, Sp2GNO across all relevant geometric benchmarks.
- Added WNO, CNO on a suite of ODE/PDE benchmarks (e.g., Driven Pendulum, Duffing Oscillator) for regular grid tasks.

This broadens the experimental scope and covers major architectural families (spectral, graph-based, transformer-based), confirming GLNO's performance.

---

> ### Author Response · Authors · 2025-12-03
> **Summary (2/2)**
>
> 3). **Strengthened theoretical analysis** on learning and convergence are added in the appendices:
> - Computational Complexity (Appendix B.5)
> - Approximation Bounds (Appendix B.6)
> - Sample Complexity (Appendix B.7)
>
> These additions directly address the reviewer's request for convergence-oriented theory.
>
> **Current status:** The reviewer last commented: *“The paper has been strengthened with the additional examples and benchmarks…”* The final request for theoretical analysis has been fully addressed. There is no remaining concerns.
>
> ### **C. Reviewer tuq4**
>
> **Initial concerns**: Comparison between spectral and non-spectral operators; More challenging problems and more baselines; Mathematical background for ML readers; Unclear tuning protocol.
>
> **Our revision:**
>
> 1). **Expanded discussion** on the advantages of spectral methods for global dependency modeling, with GLNO further enabling non-periodic dynamics and geometric alignment.
>
> 2). **Added baseline:**
> - Added ODE/PDE tasks (e.g., Lorenz system, beam equation) and a wider set of baselines across architectural families
> - Clarified our baseline selection rationale in the Experiments section.
>
> 3). **Improved mathematical accessibility:** Expanded the appendix with backgrounds on the pole-residue forms, the property of the Laplace transform, making the paper more accessible to broader readers.
>
> 4). **Clarified baseline tuning:**
> Clarified the unified tuning strategy applied across all baselines to ensure fairness.
>
> **Current status:** No further objections were raised during rebuttal.
>
> ### **D. Reviewer gWyQ**
> **Initial concerns**: Theoretical rigor; Limited experimental scope; Weak interpretability
>
> **Our revision:**
>
> 1). **Strengthened theoretical rigor:**
>
> - Provided complete derivations, including the geometric Laplace transform and the pole-residue formulation in the appendix.
> - Restructured the theoretical section (4.2) for clarity.
>
> 2). **Expanded experimental Scope:**
>
> - Expanded baselines to include Geo‑FNO, GINO, LSM, GKNO, Sp2GNO, etc.
> - Highlighted ablation studies (GLNO w/o σ).
>
> 3). **Improved interpretability:**
>
> - Enhanced the theoretical interpretability by adding analyses of the aperiodic basis properties and signal decomposition mechanisms.
> - Explicitly connected theoretical capabilities (e.g., transient modeling, geometric invariance) to observed performance gains on various benchmarks.
>
> **Current status:** The reivewer commented *“Thanks… I have adjusted the score.”* No further objections were raised during rebuttal.
>
> ## 3. Contribution summary
>
> Across reviewer interactions, the paper now clearly presents substantial improvements:
>
> **Theoretical contributions**
> - **New operator class**: GLNO integrates aperiodic Laplace-domain spectral modeling with intrinsic manifold geometry, enabling unified treatment of transient and steady-state dynamics on arbitrary Riemannian manifolds.
> - **Complete mathematical foundation**: The generalized Laplace basis, geometric Laplace transform, and spectral convolution structure are fully derived and coherently presented.
> - **Learning-theoretic analysis**: Computational complexity, approximation bounds, and sample-complexity estimates provide principled insight into expressivity, convergence behavior, and data requirements.
>
> **Methodological contributions**
> - **Unified transient–geometric modeling**: Learnable exponential bases combined with Laplace–Beltrami eigenstructure yield a principled, mesh-invariant operator applicable across irregular domains.
> - **Efficient implementation**: GLNO maintains a competitive per-epoch training cost, with LBO eigen-decomposition required only once per mesh.
> - **Improved interpretability**: The paper now clearly links theoretical properties (transient handling, geometric invariance, parameter efficiency) to observed empirical behavior.
>
> **Empirical contributions**
> - **Broader benchmarks**: Covers ODE/PDE dynamics (e.g., Duffing, Lorenz, beam equation) and complex real-world manifolds (RNA, human surfaces, SHREC-11).
> - **Comprehensive baselines**: includes CNO, WNO, Geo-FNO, GINO, GKNO, Sp2GNO, Transolver, LSM, covering Fourier-, graph-, transformer-, wavelet-, and latent spectral families.
> - **Consistent performance**: GLNO achieves state-of-the-art or competitive results across both Euclidean and non-Euclidean tasks; ablations confirm the importance of the aperiodic basis.
>
> ---
>
> The revised paper now presents a coherent, theoretically grounded, and extensively validated contribution. No reviewer raised remaining technical objections as of the final comment, and multiple reviewers explicitly indicated that the paper is now appropriate for acceptance.
>
> We hope that this summary provides a basis for your recommendation. We sincerely appreciate the significant effort invested by the Committee, especially under the exceptional circumstances of this year.
>
> Sincerely,
>
> The Authors

---

### Meta-Review · Area_Chair_s38T · 2026-01-06

**Summary:**

The paper proposes Geometric Laplace Neural Operator (GLNO) as an alternative to Fourier-style operators for i) aperiodic dynamics and ii) learning on irregular geometries. The core idea is to replace purely oscillatory bases with a Laplace-style basis that includes an exponential envelope, $e^{-zx}=e^{-\sigma x}e^{-i\omega x}$, and then lift this construction to manifolds using Laplace–Beltrami eigenfunctions and an additional geometric factor.  Because learned poles are continuous while manifold spectra are discrete, the method uses a Gaussian filtering/projection step to map poles onto discrete eigenfrequencies.

The rebuttal improved the submission substantially. It expanded baselines, corrected inconsistencies, and added additional theory material.

Even after the discussion, I do not think the paper reach the bar for acceptance, mainly because key claims still rest on assumptions that are not cleanly scoped or validated in experiments, and the evaluation and positioning remains incomplete in a few important places.

**Reviewer Concerns:**

Concerns that were largely addressed in the rebuttal/discussion:
Reviewer Fu1v asked for clearer presentation, stronger geometry baselines, and complexity discussion; these were addressed in the rebuttal. Reviewer gWyQ raised issues about rigor, inconsistencies, and missing ablations; the authors responded with derivation/notation fixes and additional experiments. Reviewers YJMB and tuq4 requested broader benchmarking and more background. The rebuttal added material, and these points were at least partially addressed.

Concerns that remain outstanding:
First, the paper’s “dynamic mesh” narrative is still not supported by experiments. The response suggests recomputing LBO eigenpairs over time, but also acknowledges the associated preprocessing cost and does not provide a convincing dynamic mesh study.
Second, several key steps in the geometric Laplace story rely on strong structural assumptions (e.g., homogeneity/radial symmetry or slowly varying geometric factors). These may be reasonable in a subset of settings, but the manuscript still reads more generally than what is actually justified, and the scope statements need to be more precise.

**Reviewer Scores:**

Based on the discussion, Fu1v increased to 6 and gWyQ likey increased to 4, while YJMB and tuq4 likely stayed near their original scores since their remaining concerns are only partially resolved.

---

### Decision · Program_Chairs · 2026-01-26

Reject